# Towards Data-Agnostic Pruning At Initialization: What Makes a Good Sparse Mask?

**Hoang Pham**[1], **The-Anh Ta**[2], **Shiwei Liu**[3,6], **Lichuan Xiang**[4], **Dung D. Le**[5],
**Hongkai Wen**[4], **Long Tran-Thanh**[4]
[1] FPT Software AI Center, [2] CSIRO's Data61, [3] University of Texas at Austin,
[4] University of Warwick, [5] VinUniversity [6] Eindhoven University of Technology
hoang.pv1602@gmail.com, theanh.ta@csiro.au,
{L.Xiang.2, Hongkai.Wen, Long.Tran-Thanh}@warwick.ac.uk
dung.ld@vinuni.edu.vn, shiwei.liu@austin.utexas.edu

## Abstract

Pruning at initialization (PaI) aims to remove weights of neural networks before training in pursuit of training efficiency besides the inference. While off-the-shelf PaI methods manage to find trainable subnetworks that outperform random pruning, their performance in terms of both accuracy and computational reduction is far from satisfactory compared to post-training pruning and the understanding of PaI is missing. For instance, recent studies show that existing PaI methods only able to find good layerwise sparsities not weights, as the discovered subnetworks are surprisingly resilient against layerwise random mask shuffling and weight re-initialization. In this paper, we study PaI from a brand-new perspective – the topology of subnetworks. In particular, we propose a principled framework for analyzing the performance of PaI methods with two quantities, namely, the number of effective paths and effective nodes. These quantities allow for a more comprehensive understanding of PaI methods, giving us an accurate assessment of different subnetworks at initialization. We systematically analyze the behavior of various PaI methods through our framework and observe a guiding principle for constructing effective subnetworks: *at a specific sparsity, the top-performing subnetwork always presents a good balance between the number of effective nodes and the number of effective paths.* Inspired by this observation, we present a novel data-agnostic pruning method by solving a multi-objective optimization problem. By conducting extensive experiments across different architectures and datasets, our results demonstrate that our approach outperforms state-of-the-art PaI methods while it is able to discover subnetworks that have much lower inference FLOPs (up to 3.4×). Code is available at: `https://github.com/pvh1602/NPB`.

## 1 Introduction

Deep neural networks have achieved state-of-the-art performance in a wide range of machine learning applications [3, 13, 39, 38]. However, the huge computational resource requirements limit their applications, especially in edge computing and other future smart cyber-physical systems [23, 48, 37, 47, 2]. To overcome this issue, a number of approaches have been proposed to reduce the size of deep neural networks without compromising performance, among which *pruning* has received voluminous attention [10, 24, 8]. Traditional pruning approaches mainly focus on accelerating inference, which usually require a pre-trained dense network [33, 20, 31].

As large language models (LLMs) [3, 43, 44] continue to gain popularity, endeavors start to explore the possibility to prune models before training while matching the dense performance. Lottery Ticket

37th Conference on Neural Information Processing Systems (NeurIPS 2023).

Hypothesis (LTH) [15, 4, 5] provides empirical evidence for this research goal by discovering sparse subnetworks that can be trained from scratch to the comparable performance of the dense network. However, LTH typically involves the costly iterative pruning and re-training process, whose overall cost is much more than training a dense network.

This issue raises an intriguing research question: How to identify sparse, trainable subnetworks at initialization without pre-training? Specifically, a successful pruning before training method can significantly reduce both the cost of memory and runtime, without sacrificing performance much [46]. This would make neural networks applicable even in scenarios with scarce computing resources [1, 47]. As such, many methods for PaI have been proposed [26, 42, 9, 45, 1, 29]. While these methods are based on a number of intuitions (e.g., leveraging the gradient information [26, 45]), they typically measure the importance of network parameters. Their performance in terms of both accuracy and computational reduction is far from satisfactory compared to post-training pruning and the understanding of PaI is missing. More recently, Frankle et al. [17], Su et al. [40] observe a rather surprising phenomenon: for PaI methods, layerwise shuffling connections of pruned networks does not reduce the network's performance, which suggests that layerwise sparsity ratios might be more important than weight-level importance scores. This indicates that in searching for good subnetworks at initialization, the topology of subnetworks, particularly, the number of input-output paths and active nodes, plays a vital role and should be investigated more extensively.

In this paper, we first present a **counter-argument** against the previous findings: while Frankle et al. [17] show that PaI methods are insensitive to random shuffling, we find this is not true in the extreme sparsity regime (> 99%), as the number of effective paths heavily suffers from random shuffling. In layerwise shuffling experiments (see Section 3.3), shuffling connections results in more effective nodes but substantially fewer input-output paths. In normal sparsity levels, shuffling weights in regular sparsities can maintain and even increase effective parameters [17] and the number of activated nodes [36]. This increases representation capacity and performance of the subnetwork. However, at extremely sparse levels, shuffling still preserves roughly the same number of effective nodes, but the performance of shuffled subnetworks drops significantly compared to their unshuffled counterparts. This is because random weight shuffling damages effective paths and hampers information flow within the subnetwork. These findings suggest that separately considering effective paths or nodes is insufficient to fully capture the behavior of subnetworks generated by PaI methods.

To underscore the critical importance of nodes and paths in designing network architecture, we conduct an analysis of their quantities in the context of Neural Architecture Search (NAS). For instance, Figure 1 illustrates the relationship between candidate's performance, nodes and paths in NAS-Bench-Macro [41] which has 6, 561 models (more in Section 3.4). To ensure fairness in terms of model size, we compare networks with similar number of parameters (i.e., in a range of 300k parameters in Figure 1). Remarkably, architectures exhibiting a higher number of input-output paths and active neurons concurrently yield superior performance. This highlights the crucial role of simultaneously considering both node and path in the successful design of networks at initialization.

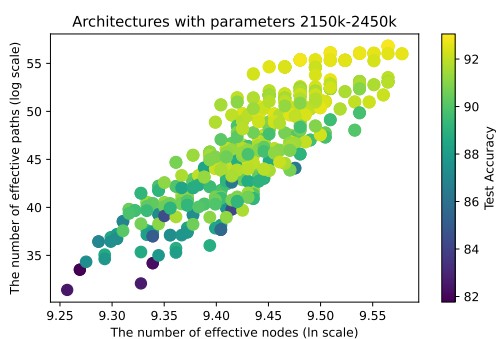

Figure 1: The accuracy of architectures in NAS-Bench-Macro benchmark along with the number of effective nodes and paths.

To improve upon the current understanding of PaI methods, we introduce a novel framework from the perspective of subnetwork topology to *provide a more accurate and comprehensive explanation of the performance of different approaches*. In particular, we propose the joint usage of both the number of input-output paths (a.k.a. effective paths) and activated (i.e., effective) nodes to interpret behaviors of different PaI methods in a more comprehensive way (see Section 3.3 for more details). Inspired by the guiding principle observed through our framework, we further introduce NPB, a novel pruning at initialization by solving a node-path balancing objective. We show that *NPB outperforms state-of-the-art PaI methods in almost all settings, while achieving much lower inference FLOPs*. In summary, our main contributions are:

- We propose a unified effect node-path perspective to understand the behavior of PaI, particularly considering metrics of effective nodes and paths as proxies for the performance of PaI methods (Section 3). We revisit the layerwise shuffling sanity check on PaI methods and provide unified explanations for their behaviors based on these metrics in a wide range of sparsities (Section 3.3).

- We discover a new relation between the proposed metrics and the performance of subnetworks, termed the Node-Path Balancing Principle, that suggests a non-trivial balance between nodes and paths is necessary for optimal performance of PaI (Section 4).

- We present a novel data-agnostic pruner, NPB, by solving a multi-objective optimization problem to give a proof-of-concept for our principle. With extensive experiments, we show that NPB outperforms the best state-of-the-art PaI methods, PHEW, on 11 out of 12 settings while ours produces more efficient subnetworks with much fewer inference FLOPs (up to 3.4×) and faster in pruning time than PHEW (up to 14.5×) (Section 5).

## 2   Related Work

**Neural Network Pruning.**   Neural network pruning methods [25, 21, 20, 12, 32] traditionally focus on pruning trained models based on pre-defined criteria, and then resulting subnetworks will be fine-tuned to converge. Recently, [15, 16, 4, 5] empirically show the existence of randomly initialized subnetworks (lottery tickets) which when trained from scratch or in early training iterations, that can achieve competitive performance with their dense counterparts. Unfortunately, finding lottery tickets is computationally expensive due to the train and prune cycle [16, 15]. Gradual pruning methods [49, 18] interleave the pruning and training, which are usually cheaper than pruning after training, but the network still needs to be trained to choose the ideal sparse subnetwork. Other methods [6, 7] apply one-shot pruning during training to further reduce the computational cost. Dynamic sparse training [30, 14, 28, 27], on the other hand, start with a (random) sparse network and update network connectivity during training. While pruning before training methods [26, 45, 36, 42, 1] determine subnetworks by the network initialization, gradient information, and network topology. However, experimental results done by Frankle et al. [17], Su et al. [40] show that current criteria of PaI methods may not be sufficient to obtain a subnetwork with good performance.

**Pruning and Network Shape.**   Since PaI methods do not utilize training data [42, 36] or use only negligible portions of data [26, 45] to obtain gradient information without training, the configuration of nodes and connections is an essential source of information for optimizing the performance of pruned networks. It turns out that some PaI methods implicitly optimize certain aspects of network shape. In particular, SynFlow [42] preserves the number of input-output paths as synaptic strength, but often creates more isolated neurons in pruned networks. The works of Patil and Dovrolis [36] and Gebhart et al. [19] aim to preserve proxies in terms of path kernels which are also directly related to the shape of subnetworks. Furthermore, while PHEW [36] additionally implements random walks to increase the number of effective nodes, it unintentionally decreases the number of input-output paths. Our new point of view on node-path balancing would be helpful to systematically optimize network configuration for better performance. Other works also consider the number of effective nodes and effective paths to capture the capacity of pruned subnetworks [45, 34] where these numbers are considered separately.

## 3   Methodology

### 3.1   Pruning at Initialization Methods

Given a $L$ layer neural network, we denote $\mathbf{w} = (w_1, \ldots, w_L)$ as the set of vectorized weights. Pruning generates binary mask vectors $m_\ell \in \{0, 1\}^{d_\ell}$ ($d_\ell$ is the number of connections in layer $\ell$) yielding sparse neural networks with sparse weights $m_\ell \odot w_\ell$ - the elementwise product of masks and weights. Sparsity is defined as the fraction of weights being removed: $s = 1 - \frac{\sum m_\ell}{\sum d_\ell} \in [0, 1]$.

A pruning method usually consists of two operations: *score* and *remove*, where *score* takes as input weights of the network and outputs an important score for each weight: $z_\ell = score(w_\ell) \in \mathbb{R}^\ell$; then *remove* takes as input the scores $\mathbf{z} = (z_1, \ldots, z_L)$ and the sparsity $s$, and outputs the masks $m_\ell$ with

overall sparsity $s$. Pruning can be done in one-shot or iteratively. For one-shot pruning, we only generate the scores once, then prune the network up to sparsity $s$. For iterative pruning, we repeat the processes of the score, then prune from sparsity $s^{(t-1)/T}$ to $s^{t/T}$ repeatedly $T$ times.

**Random.** This method assigns each connection with a random score from a uniform distribution $\mathcal{U}(0, 1)$. Random pruning empirically prunes each layer to target sparsity $s$ [29].

**SNIP.** SNIP was introduced by Lee et al. [26] with the pruning objective of reducing connection sensitivity to the training loss. One passes a mini-batch of data through the network and computes the score $\mathbf{z}$ for weight $\mathbf{w}$ of SNIP as $\mathbf{z} = |\mathbf{w} \odot \nabla_{\mathbf{w}} \mathcal{L}|$.

**Iterative SNIP.** This is an iterative variant of SNIP [9] with the same important score. But, iterative SNIP gradually prunes the remaining weights with lowest scores from sparsity $s^{\frac{t-1}{T}}$ to sparsity $s^{\frac{t}{T}}$ iteratively $T$ times for $t = 1, 2, \ldots, T$.

**SynFlow.** SynFlow [42] is an iterative and data-agnostic PaI method. The pruning objective of SynFlow is to make the network remains connected until the most extreme possible sparsity. The weight scores are computed as follows. One first replaces all weights in the network by their absolute values. Then, an $\mathbf{1}$ input tensor is passed through the network, and one computes the sum of the logits as $R = \mathbf{1}^{\top}(\prod_{\ell=1}^{L} |w_{\ell}|)\mathbf{1}$. Finally, the score of weight $\mathbf{w}$ is computed as $\mathbf{z} = |\mathbf{w} \odot \nabla_{\mathbf{w}} R|$. SynFlow prunes the network iteratively $T$ times.

**PHEW.** PHEW [36] is also an iterative and data-independent PaI method. It selects a set of input-output paths to be preserved. These paths are chosen through random walks, biased towards higher-weight magnitudes. The selection starts with a unit that is selected through round robin procedure. This process continues until the subnetwork achieves the predefined sparsity.

**ERK.** Erdős-Rényi (ER) first used by Mocanu et al. [30] to sparsify Multilayer Perceptron (MLP) networks using a random topology that allocates higher sparsity to larger layers. Evci et al. [14] extends ER to a convolutional version called Erdős-Rényi-Kernel (ERK) which scales the sparsity of the convolutional layer in proportion to the number of neurons/channels in a layer.

## 3.2 Metric Definition

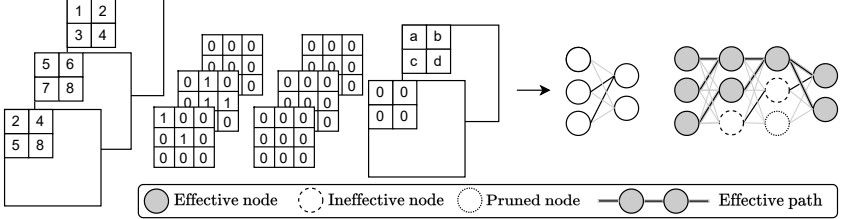

Figure 2: An example of effective paths and effective nodes.

In a sparse network, it is intuitively clear that one should arrange the connections into a configuration neither too thin nor too spread-out to have good information propagation during training. For a better measurement, we propose using two metrics to evaluate the quality of subnetworks. Please refer to Appendix B for detailed discussions and Python code for calculating the metrics.

**Effective Path.** We define a path to be *effective* if it connects an input node to an output node without interruption (see Figure 2). Metrics based on paths are mentioned in [42, 19] as $l_1$ and $l_2$ path norms, respectively. In this paper, we only take into account the number of paths.

**Effective Node/Channel.** A node/channel is effective if at least one effective path goes through it (demonstrated as the right part in Figure 2). This concept is also considered in [36, 17]. For convolutional layers, we consider a kernel as a connection, and a channel as a node, and then convert the convolutional layer into a fully connected layer (see the left part in Figure 2).

## 3.3 Layerwise Shuffling Phenomenon

In this section, we investigate the intriguing phenomenon that layer-wise reshuffling the subnetwork found by PaI methods still produces competitive accuracy [17, 40]. Based on metrics, we provide a

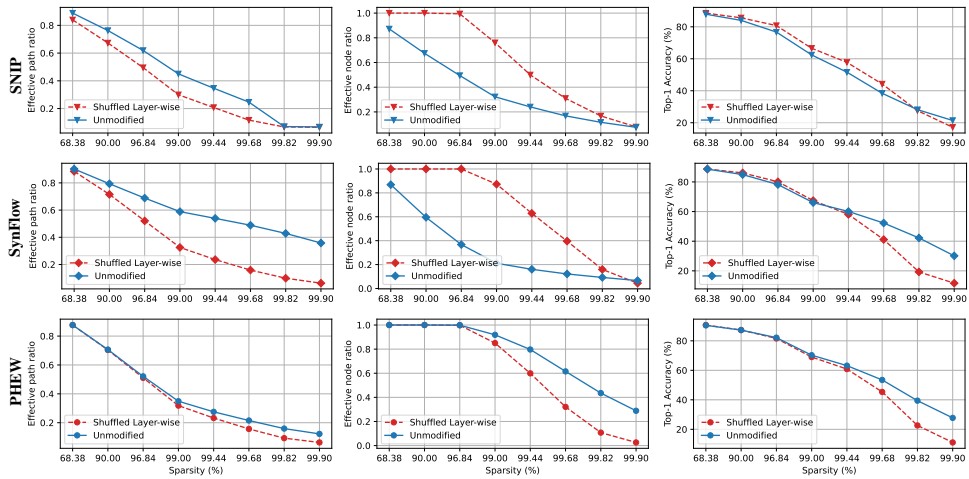

Figure 3: Layerwise shuffling results on various sparse subnetworks of ResNet20 produced by SNIP, SynFlow, and PHEW on CIFAR-10.

new way to understand why reshuffling subnetworks work and when they fail. We first use three PaI methods, i.e., SNIP [26], SynFlow [42], and PHEW [36], to find the subnetworks. Then, we randomly shuffle the pruning mask $m_l$. All the subnetworks (both unmodified and shuffled) are trained in the same setting. Finally, we compute the ratio of the number of active paths and nodes after and before pruning and visualize the average scores in Figure 3.

In SNIP and SynFlow, the number of effective nodes drops significantly as sparsity increases (blue lines in second columns in Figure 3). After reshuffling, the connections are distributed uniformly in each layer leading to wider subnetworks while the number of effective paths decreases. In contrast, PHEW focuses on increasing the number of effective nodes by gradually adding new paths such that the network is as wide as possible. Consequently, reshuffling hurts the concrete network configuration and then reduces both the number of effective nodes and effective paths as sparsity is higher.

At sparsity levels below 99%, layerwise shuffled subnetworks demonstrate competitive performance or even better than unmodified counterparts, especially with SynFlow and SNIP. In addition to effective connection preservation as discussed in prior work [17], *the representation capacity of shuffled sparse networks is enhanced with SynFlow and SNIP, attributed to the increase in the number of effective nodes* [35] while maintaining considerable input-output paths. In more details, Figure 3 shows that these two metrics for SynFlow and SNIP shuffled versions closely resemble to unmodified subnetworks of PHEW with corresponding sparsities.

However, when the network becomes more sparse, the number of input-output paths decreases substantially (along with the reduction of effective parameters see Appendix D). Even though layerwisely shuffled networks become wider, *the limited number of effective paths damages the information flows in the subnetworks*. These explain why the accuracy of shuffled subnetworks is reduced significantly compared to the unmodified ones in intensive sparsities.

These observations indicate that increasing the number of active paths or nodes alone might not be sufficient in the design of PaI methods. We hypothesize that to have good subnetworks, the number of effective paths and effective nodes should be concurrently considered. If we balance these two metrics well, the performance after the training of subnetworks will be enhanced.

### 3.4 NAS Observations

Our research delves into the fundamental aspects of network architecture design by investigating the interplay between nodes and paths. Going beyond the realm of pruning literature, we conduct an analysis of node and path metrics in the NAS benchmark. It is noteworthy that networks in NAS are dense networks. In particular, we focus on NAS-Bench-Macro [41], a macro benchmark comprising 8 searching layers. Each layer offers three candidate blocks, resulting in a staggering 6,561 distinct networks with parameter counts ranging from 300k to 3M. To ensure a fair comparison

among candidates, we specifically consider networks with similar parameter counts within the 300k range. We compute metrics across four different parameter ranges and visualize them in Figure 4.

In the context of NAS, it's important to consider that other aspects beyond node-path balance can contribute to final classification accuracy. For instance, in the case of NAS-Bench-Macro, networks with similar node-path balance but varying classification accuracies can be affected by other architectural configurations, such as differences in pooling layers, kernel sizes, and expansion ratios in the MobileNet-v2 block. These architectural variances often result in different numbers of parameters, influencing the overall network performance. However, in the sparse neural network context, pruning methods focus on maintaining the same network structure while pruning connections within the network based on specific sparsities. Consequently, our experiments did not explore other architectural elements beyond node and path balance. In general,

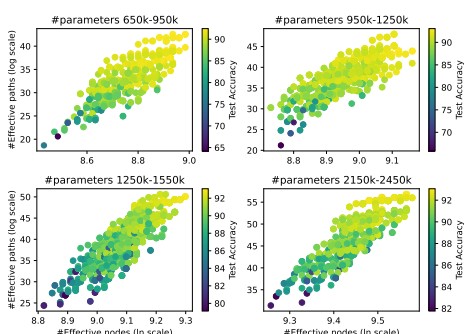

Figure 4: The accuracy of network candidates in NAS-Bench-Macro benchmark along with the number of activated nodes and paths in different parameter ranges.

we can see that networks with larger numbers of nodes and paths tend to exhibit higher performance. Our findings unveil a strong correlation between the two metrics (node, path) and the final performance of network candidates, both playing pivotal roles in achieving superior performance. This highlights the critical significance of both nodes and paths in the initial stages of designing subnetworks that yield exceptional results.

# 4 Node-Path Balancing Principle

## 4.1 Node-Path Balancing Principle

From observations in Section 3.3 and NAS observations in Section 3.4, both effective paths and nodes have shown their critical roles in the performance of subnetworks. We now formally state the *Node-Path Balancing Principle* (NPB): The combination of both the numbers of effective nodes and effective paths is a proxy for the potential performance of subnetworks under pruning. A pruning at initialization method which produces pruned subnetworks with too many effective paths (resp. effective nodes) will have less than necessary the number of effective nodes (resp. effective paths) and consequentially has suboptimal performance. It is necessary to balance the number of effective nodes and effective paths for better performance of pruned subnetworks.

## 4.2 Proposed Method: NPB

Building upon the aforementioned principle, we elegantly transform the pruning problem into a multi-objective optimization problem. Specifically, our objective is to maximize both the number of effective nodes and effective paths concurrently given an architecture and desired sparsity ratio. We formulate this intriguing problem as follows:

Given an architecture $A$ with parameter $\mathbf{W} \in \mathbb{R}^N$ where $N$ is the total number of parameters and sparsity ratio $s$. Denote $f_p$ as the total number of input-output paths, $f_n$ as the number of activated nodes, and consider the mask for parameter $\mathbf{M} = \{0, 1\}^N$ as variable to solve.

$$\underset{\mathbf{M}}{\text{Maximize}} \quad \alpha f_n + (1 - \alpha) f_p$$
$$s.t \quad \|\mathbf{M}\|_1 \leq N(1 - s)$$

where $\alpha$ is a coefficient. Solving node-path balancing objective globally over the whole neural networks is a non-trivial problem. We sidestep this challenging issue by solving a sequence of easy problems to obtain good approximated solutions. In particular, we propose an approximation for solving this problem by doing layer by layer through convex optimization. The approximation problem is solved efficiently via the available convex optimization library.

To be simple, we consider a linear layer as an example, in which, $v$ denotes the node, $l$ is the layer index, $\mathbf{m}^{(l)} \in \mathbb{R}^{h^{(l)} \times h^{(l+1)}}$ is the mask for layer $l$ in which $h^{(l)}$ the number of neurons in layer $l$. We denote $P(v_i^{(l)})$ as the number of path go to node $v_i$ in layer $l$ (e.g., at input layer $P(v_i^{(0)}) = 1$). We formulate the pruning problem in layer $l$ as an optimization problem in which we maximize the number of paths from input to layer $l+1$ $f_p^{(l+1)}$ and the total activated nodes $f_n^{(l+1)}$ in two layers $l$ and $l+1$. For brevity, we denote them as $f_p$ and $f_n$. We set $\mathbf{m}^{(l)}$ as:

$$m_{ij}^{(l)} = \begin{cases} 0 & \text{if pruned} \\ 1 & \text{if not pruned} \end{cases}$$

The number of paths to $v_j^{(l+1)}$ in layer $l+1$ is sum of all paths to nodes in layer $l$ connecting to $v_j^{(l+1)}$

$$P(v_j^{(l+1)}) = \sum_i^{h^{(l)}} m_{ij}^{(l)} P(v_i^{(l)})$$

Then the total number of effective paths to layer $l + 1$ is:

$$f_p(m) = \sum_j^{h^{(l+1)}} P(v_j^{(l+1)}) \tag{1}$$

A node $v_i$ in layer $l$ is activated if and only if there are paths pass through it and edges connect from it to nodes in layer $l + 1$, which is formulated as below:

$$P(v_i^{(l)}) \sum_j^{h^{(l+1)}} m_{ij}^{(l)} \geq 1 \Leftrightarrow \min(P(v_i^{(l)}) \sum_j^{h^{(l+1)}} m_{ij}^{(l)}; 1) = 1$$

And a node $v_j$ in layer $l + 1$ is effective when there exist attached nodes in layer $l$ that connect to it,

$$\sum_i^{h^{(l)}} m_{ij}^{(l)} P(v_i^{(l)}) \geq 1 \Leftrightarrow \min(\sum_i^{h^{(l)}} m_{ij}^{(l)} P(v_i^{(l)}); 1) = 1$$

The layer-wise objective for the node becomes:

$$f_n(m) = \sum_i^{h^{(l)}} \min(P(v_i^{(l)}) \sum_j^{h^{(l+1)}} m_{ij}^{(l)}; 1) + \sum_j^{h^{(l+1)}} \min(\sum_i^{h^{(l)}} m_{ij}^{(l)} P(v_i^{(l)}); 1) \tag{2}$$

In convolution layers with a kernel of height $h$ and width $w$, we let the variable $m_{ij}$ have the value from 0 to $hw$ representing this kernel. After solving $\mathbf{m}$, in each kernel, we assign $|m_{ij}|$ parameters to entries whose initialized weights have the highest magnitude.

Along with two objectives, we also consider a regularization term which aims to encourage activating as many kernels as possible in each layer. Besides, since we optimize the node and path per layer, the solution will not be the global one. We can view the regularizer as an adjustment term, which moves the local solution around to figure out the better ones.

$$R = \sum_i \sum_j \min(m_{ij}^{(l)} - 1; 0) \tag{3}$$

From Equations 1, 2, and 3 the final objective becomes:

$$\underset{\mathbf{m}^{(l)}}{\text{Maximize}} \quad \alpha f_n + (1 - \alpha) f_p + \beta R \tag{4}$$

$$s.t \quad \|\mathbf{m}^{(l)}\|_1 \leq N^{(l)}(1 - s^{(l)}) \tag{5}$$

where $\alpha$ is a coefficient to control the balance, $\beta$ is a hyperparameter, and the constraint is the number of unpruned parameters that satisfies the sparsity. To ensure the same scale between elements in the main objective function, we normalize the number of paths, nodes, and the regularizer in each layer with their maximum possible values, respectively. Note that, optimizing nodes is much easier than paths so we select small values of alpha which can be prior knowledge. In particular, in Section 5, we fix $\alpha = 0.01$ and $\beta = 1$ to all settings. We optimize [1] the network structure sequentially layer by layer from input to output in which the layer-wise sparsity level is found by ERK method [14, 29]. We describe our method in Algorithm 1 and the pseudo code for optimizer in Appendix C.

---

[1] We use the default mixed integer programming solver in CVXPY library

**Algorithm 1** Node-Path Balancing Pruner

---

1: **Inputs:** Final sparsity $s$, weights $\mathbf{w}_0 = \mathbf{1}$, balancing coefficient $\alpha$, and hyperparameter $\beta$
2: Obtain layer-wise sparsity $s^{(l)}$ by using ERK method
3: Define unit input $\mathbf{x} = \mathbf{1} \Rightarrow P^{(0)} = \mathbf{1}$
4: **for** $l = 0, \ldots, L$ **do**
5:     $\mathbf{m}^{(l)} \leftarrow optimize(\mathbf{m}^{(l)}, s^{(l)}, P^{(l)})$ # solve Eq.4
6:     Set $\mathbf{w}_0^{(l)} = \mathbf{m}^{(l)}$
7:     Extract layer $l + 1$ output when put $\mathbf{x}$ to the network with $\mathbf{w}_0$ to obtain $P^{(l+1)}$
8: **end for**
9: **Return:** Mask $\mathbf{M}$

---

## 5 Evaluation

### 5.1 Experimental Settings

We conduct experiments on three standard datasets: CIFAR-10, CIFAR-100, and Tiny-Imagenet. Following [42], we use ResNet-20 for CIFAR-10, VGG-19 for CIFAR-100, and a variant ResNet-18 with 18 layers for Tiny-Imagenet. We treat weights of all convolutional and linear layers as prunable, but we do not prune biases and weights in batch normalization layers. We run five seeds with experiments on CIFAR-10, CIFAR-100, and three seeds on Tiny-Imagenet. The average results are used for visualizations. Since the number of active paths is orders of magnitude larger than that of active nodes, and both numbers are of exponential scales, we take logarithm base 10 of the number of active paths, and logarithm base $e$ (natural logarithm) of the number of active nodes to visualize the results. To ensure the fair comparison between NPB and other baselines, we fix $\alpha = 0.01$ and $\beta = 1$ in NPB for all settings. More details on our experimental setting are in Appendix A.

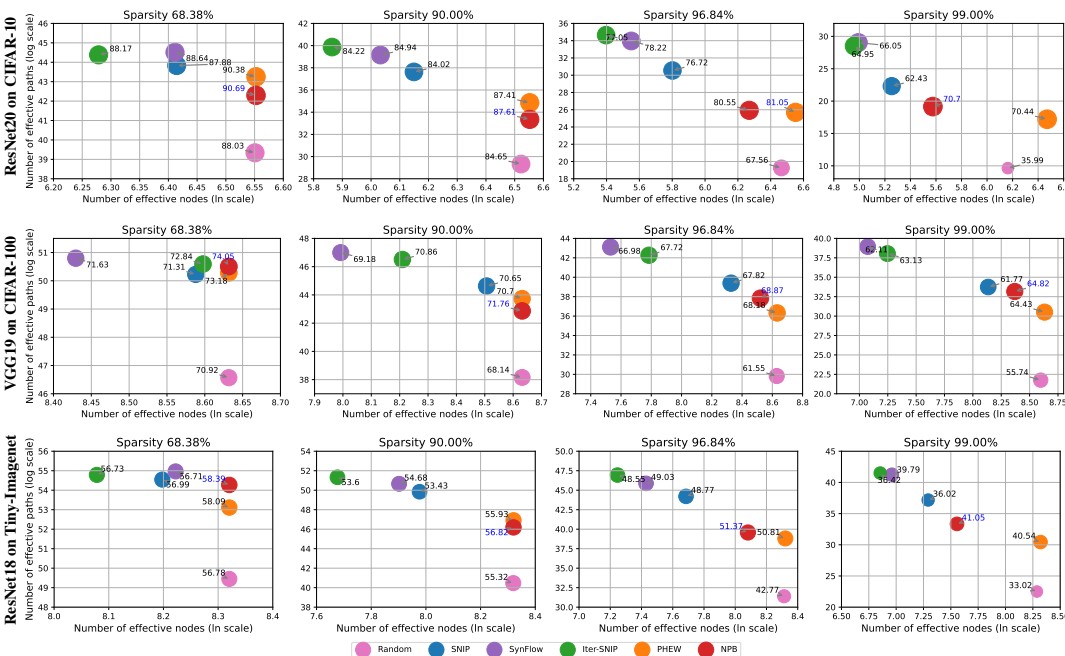

Figure 5: The number of effective paths (log scale), nodes (ln scale), and the corresponding accuracy of different PaI methods on three datasets in different sparsity levels. Best accuracy is in blue.

### 5.2 Comparison with PaI Methods

Figure 5 shows the experimental results for different PaI methods: Random, SNIP, Iter-SNIP, SynFlow, PHEW, and our NPB, with different sparsities on three settings: VGG-19 on CIFAR-100, ResNet-20 on CIFAR-10, and ResNet-18 on Tiny-Imagenet. Overall, our method NPB shows significant improvements on standard PaI methods, both data-dependent like Iter-SNIP, SNIP (improve up to

14%) and data-independent like SynFlow (increase up to 7%) in all three settings. *Notably, NPB achieves better results than state-of-the-art PaI method PHEW in almost all settings (11 out of 12 settings). Importantly, we set the same alpha and beta hyperparameters for NPB across all three settings, thereby eliminating the need for extensive hyperparameter search.* This demonstrates the robustness and generalizability of our method, making it easily applicable to various settings with consistently high performance. Ablation studies on alpha and beta can be found in Appendix G.

We discuss the experimental results from the point of view of the Node-Path Balancing principle and network shape optimization. In Figure 5, SynFlow and Iter-SNIP produce sparse networks with large number of input-output paths, which results from iterative pruning procedures. These methods gradually remove nodes of low connection degree, which makes the subnetworks become narrower while maintaining the high number of paths. As a consequence of a very high number of input-output paths, the width of resulting subnetworks is significantly reduced, limiting the representation capacity of the subnetworks [35, 36], leading to suboptimality.

Random pruning is a PaI method which produces subnetworks with the very high number of active nodes. On average, it distributes parameters uniformly to all layers and kernels, creating subnetworks of large width yet a low number of active paths. At sparsities below 90%, with an adequate number of input-output paths, subnetworks of Random pruning still perform well and are even competitive with more sophisticated PaI methods. This phenomenon is exhibited clearly in experiments on the more complex dataset Tiny-Imagenet. However, when sparsity increases over 90%, Random pruning creates a huge number of ineffective paths and parameters. Consequently, the performance of subnetworks drops significantly. We again observe the necessity to balance the optimization of the number of nodes and paths.

In Figure 5, our method NPB and PHEW produce sparse networks which, in visualization, lie in specific areas which have higher paths than Random and greater nodes than SynFlow, or Iter-SNIP. We call these are *balancing regions*. Subnetworks in these regions have broader widths and considerably many paths, which provides good representation capacity and preserves the information flow of the network. These two factors together contribute to better training of pruned subnetworks leading to superior performance. This illustrates the effectiveness of our principle.

One drawback of one-shot pruning methods like SNIP is creating a large number of ineffective parameters, which decreases the capacity and performance of subnetworks. Although our method is also one-shot, it efficiently distributes the parameters by direct optimization. Consequently, NPB enjoys a better node-path balancing level and better performance (up to 14%) compared to SNIP.

Results of all settings in Figure 5 provide evidence to support our Node-Path Balancing principle and the existence of a specific balancing region between nodes and paths at given sparsity levels. Particularly, we posit that when the number of active neurons and input-output paths of a sparse network fall within this balancing range, it will probably have a good performance after training.

## 5.3   Pruning Time and FLOPs Reduction

Table 1: Accuracy, pruning time (in seconds) and FLOPs of subnetworks for different pruning methods and compression ratios on Resnet18 - Tiny-ImageNet.

| | Accuracy (%) | | | | Pruning time (seconds) | | | | FLOPs ($10^8$) | | | |
|---|---|---|---|---|---|---|---|---|---|---|---|---|
| Sparsity (%) | 68.38 | 90.00 | 96.84 | 99.00 | 68.38 | 90.00 | 96.84 | 99.00 | 68.38 | 90.00 | 96.84 | 99.00 |
| SNIP | 56.99 | 53.43 | 48.77 | 36.02 | 5.14 | 4.95 | 5.55 | 5.64 | 11.35 | 5.77 | 3.04 | 1.55 |
| Iter-SNIP | 56.73 | 53.60 | 48.55 | 36.42 | 229.16 | 235.34 | 233.19 | 231.23 | 10.73 | 7.05 | 3.98 | 1.97 |
| SynFlow | 56.71 | 54.68 | 49.03 | 39.79 | 108.17 | 96.18 | 91.15 | 92.60 | 14.71 | 8.91 | 4.24 | 1.50 |
| PHEW | 58.09 | 55.93 | 50.81 | 40.54 | 5511.20 | 1342.03 | 471.23 | 324.78 | 14.29 | 8.35 | 3.92 | 1.50 |
| NPB | **58.39** | **56.82** | **51.37** | **41.05** | 380.52 | 375.65 | 384.32 | 387.89 | 14.37 | 5.21 | 1.74 | 0.59 |

Since NPB optimizes layer by layer, the complexity of NPB depends on network architecture (i.e., the number of layers and the size of each layer). With large layers, we sidestep the time-consuming issues by making a further step where we divide the layer into chunks with the same sparsity constraint. In particular, we split nodes in the next layer into equal parts while the input node is fixed. Then, optimizing chunks in a layer can be solved in parallel. Thank to the available convex optimization libraries we can find subnetworks more efficiently and quickly. We have computed the pruning time of our proposed method and compared it with other PaI methods in Table 1. Our pruning time is not

significantly slow compared to those iterative approaches (e.g., Iter-SNIP, SynFlow) while it is much faster (up to 14.5×) than PHEW in lower sparsity levels.

Besides pruning time, we find that the FLOPs reduction of subnetwork after pruning is more important in the context of pruning before training. We have measured FLOPs of subnetworks produced by different methods in Table 1. The result indicates that our NPB can produce subnetworks with lower FLOPs than other baselines while NPB outperforms PaI methods. More details are in Appendix E.

## 6 Conclusion

In this paper, we present a novel framework for studying PaI methods that systematically employs the configuration of pruned subnetworks based on two different metrics: the number of effective paths and the number of effective nodes. Through our framework, we discover a new relationship between these metrics, called the Node-Path Balancing Principle, which provides guidance for understanding and optimizing PaI methods. Our framework offers unified explanations for the intriguing layerwise connection reshuffling phenomenon [40, 17] of subnetworks produced by PaI methods in normal pruning sparsity regime, as well as the failure of this phenomenon in extreme sparsity levels. Furthermore, we propose a novel pruning method based on optimization approach, namely NPB, that demonstrates the potential of balancing the numbers of effective paths and nodes to improve the performance of PaI methods. Extensive experiments conducted on various model architectures and datasets show that NPB outperforms the best baseline, PHEW, on almost settings while ours produces more efficient subnetworks with much fewer inference FLOPs and faster in pruning time then PHEW. Our new perspective on the configuration of subnetworks, in terms of effective nodes and effective paths, provides new insights into the working mechanism of PaI methods and opens new research directions on neural network pruning methods, as well as designs of sparse neural network.

## Acknowledgement

Part of this work was done while Hoang Pham was visiting University of Warwick. Dung D. Le was funded by Vingroup Innovation Foundation (VINIF) under project code VINIF.2022.DA00087.

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

# A Experiment details

In this section, along with pruning at initialization (PaI) methods in the main text, we provide experimental results with GraSP [45]. In particular, GraSP is another gradient-based pruning PaI that aims to preserve the gradient flow of sparse networks obtained by pruning. The score $z$ of weight $w$ in GraSP is computed as $\mathbf{z} = -\mathbf{w} \odot (\mathbf{H}\nabla_{\mathbf{w}}\mathcal{L})$, where $\mathbf{H}$ is the Hessian of the training loss after passing a mini-batch of data through the network.

We describe our experiment settings on architectures and datasets. We use Pytorch [2] library and conduct experiments on a single GTX 3090Ti or A100 (depend on their available). We adapt from Tanaka et al. [42] source code [3] for SNIP, GraSP, SynFlow, and Random, and change the prune epochs to 100 for Iterative SNIP instead of 1 in SNIP. And we use the official code [4] of Patil and Dovrolis [36] for PHEW. For NPB method in the main experiments, we set $\alpha = 0.01$ and $\beta = 1$ for all settings.

**Datasets.** Our main experiments are conducted with CIFAR-10, CIFAR-100, and Tiny-Imagenet datasets, where:

- CIFAR-10 is augmented by normalizing per-channel, randomly flipping horizontally.
- CIFAR-100 is augmented by normalizing per-channel, randomly flipping horizontally.
- Tiny-ImageNet is augmented by normalizing per channel, cropping to 64x64, and randomly flipping horizontally.

**Architectures.** We use three different networks:

- VGG-19 is a CIFAR-100 network used in SynFlow [42]. We choose a batch-normalization version.
- ResNet-20 is a 20-layer CIFAR-10 version of ResNet created by [22]. This version has added batch normalization layers before each activation function.
- ResNet-18 is a ImageNet version with 18 layers adapted from SynFlow [42]. The first convolution has kernel size 3x3 (instead of 7x7) and max-pooling layer that follows has been removed.

We treat all of the weights from convolutional and linear layers of these networks are prunable parameters, but we do not prune the biases nor the weights in the batch normalization layers. The weights in convolutional and linear layers are initialized with Kaiming normal, while biases are initialized to be zero. We run five seeds with CIFAR experiments and three seeds with experiments on Tiny-Imagenet.

**Training details** With Iterative pruning methods SNIP and SynFlow, we use 100 pruning epochs. With methods using training data like SNIP, GraSP, and Iterative-SNIP, we randomly select 10 samples for each class, particularly, 100 data points for CIFAR-10, 1000 data points for CIFAR-100, 2000 data samples for Tiny-ImageNet. Other hyperparameters are chosen as follow:

Table 2: Summary of the architectures, datasets, and hyperparameters used in experiments.

| Network | Dataset | Epochs | Batch | Optimizer | Momentum | LR | LR Drop, Epoch | Weight Decay |
|---------|---------|--------|-------|-----------|----------|-----|----------------|--------------|
| VGG-19 | CIFAR-100 | 160 | 128 | SGD | 0.9 | 0.1 | 10x, [60,120] | 0.0001 |
| ResNet-20 | CIFAR-10 | 160 | 128 | SGD | 0.9 | 0.1 | 10x, [60,120] | 0.0001 |
| ResNet-18 | Tiny-ImageNet | 100 | 128 | SGD | 0.9 | 0.01 | 10x, [30,60,80] | 0.0001 |

---

[2]https://pytorch.org
[3]https://github.com/ganguli-lab/Synaptic-Flow
[4]https://github.com/ShreyasMalakarjunPatil/PHEW

# B  Effective metrics calculation

**Effective path.**    To exactly compute the number of effective paths, we remove the batch normalization layers, we initialize all the remaining parameters to 1. Then, we put the input vector one to the network, and the number of effective paths is the sum of logits on the output layer $R = \mathbf{1}^\top (\prod_{\ell=1}^{L} |w_\ell|)\mathbf{1}$.

More precisely, we face problems with pooling layers in convolutional neural networks. With max pooling layer, we simply do not modify the output of this layer. At that time, the result is the maximum number of paths in subnetworks. With average pooling layer, since all inputs of this layer contribute to the output, we change the average operator to the sum operator to exactly compute the number of effective paths. We all use ReLU activation functions in computing this metric since this function does not affect the results of calculations.

**Effective parameter.**    We follow [17] when identifying which is effective parameter. Similar to computing effective paths, we make further steps. After having the sum of logits, we compute the gradients of this sum with respect to weights $\nabla_\mathbf{w} R$. Then, if an unpruned weight has a non-zero gradient, it is effective and vice versa. Effective parameters are dense edges that connect two effective nodes as visualized in Figure 2.

**Effective node/channel.**    With fully connected layers, if all connections to one node or out of one node are pruned, this node is pruned node. If there exist connections to a node but all of these connections are ineffective, then this node becomes ineffective In convolutional layers, instead of nodes, we have channels. We consider a kernel as a connection, a channel as a node, and then convert the convolutional layer into a fully connected layer. The connection is pruned if and only if all parameters in the corresponding kernel are removed. Finally, identifying the effective nodes/channels is similar to the way in fully connected layers.

```python
def metric_calculation(model, mask):
    """
    model:  network architecture
    mask:   mask for subnetwork
    """
    n_eff_paths = 0
    n_eff_nodes = 0
    n_eff_params = 0

    # Initialize network with pruned weight = 0 and kept weight = 1
    for name, param in model.named_parameters():
        param.copy_(mask[name])

    x = torch.ones((1,c,h,w)) # c: channel - h: height - w: width
    y = model(x)
    sum_logits = y.sum()
    n_eff_paths = sum_logits.item()
    sum_logits.backward()
    with torch.no_grad():
        for name, param in model.named_parameters():
            eff_param = torch.where(param.grad.data!=0, 1, 0)
            n_eff_params += torch.sum(eff_param)

            eff_in_node = torch.where(torch.sum(eff_param,d=0)>0, 1, 0)
            n_eff_nodes += torch.sum(eff_in_node)
        # with output layer
        eff_out_node = torch.where(y>0, 1, 0)
        n_eff_nodes += torch.sum(eff_out_node)

    return n_eff_paths, n_eff_nodes, n_eff_params
```

Listing 1: Metric calculation example in fully connected neural networks

# C Layer-wise Mask Optimization

We provide the pseudo code solving the mask optimization problem based on the cvxpy library [11].
We use the default solver provided in this tool.

```python
import math
import cvxpy as cp
import numpy as np
import torch

def optimize_layerwise(mask, inp, sparsity, alpha=0.1, beta=0.1,
    max_param_per_kernel=None):
    """
    mask: mask of this layer
    inp: input of this layer
    sparsity: the sparsity of this layer
    alpha: is the balancing coefficient
    beta: is the regularizer coefficient
    max_param_per_kernel: max param in a kernel
    """

    if len(mask.shape) == 4: # Convolution layer
        C_out, C_in, K, W = mask.shape
        max_param_per_kernel = K * W
        P_in = torch.sum(inp, dim=(1,2)).numpy()
    else: # Linear layer
        C_out, C_in = mask.shape
        max_param_per_kernel = 1
        P_in = inp.numpy()

    # Params in layer
    n_params = int(math.ceil((1-sparsity)*mask.numel())) # This has to be
     integer

    # Define mask variable
    M = cp.Variable((C_in, C_out), integer=True)

    sum_in = cp.sum(M, axis=1) * P_in # P_in * \sum_{j}^{C_out} M_{ij} =>
     shape: C_in
    sum_out = cp.sum(M*P_in, axis=0) # \sum_{i}^{C_in} M_{ij} * P_in[i]
    => shape: C_out

    # If eff_node_in is small which means there is a large number of
    input effective node
    sum_eff_node_in = C_in - cp.sum(cp.pos(1 - sum_in))
    sum_eff_node_out = C_out - cp.sum(cp.pos(1 - sum_out))

    # Optimize nodes
    max_nodes = C_in + C_out
    A = (sum_eff_node_in + sum_eff_node_out) / max_nodes  # Scale to 1

    # Optimize paths
    max_path = compute_max_path(P_in, n_params, C_out,
    max_param_per_kernel)
    B = (cp.sum(P_in @ M)) / max_path

    # Regulaziration
    Reg = (n_params-cp.sum(cp.pos(1-M))) / n_params  # encourage number
    of edges

    # Constraint the total activated params
    constraint = [cp.sum(M) <= n_params, M <= max_param_per_kernel, M >=
    0]

    # Objective function
```

```
53    obj =cp.Maximize(alpha * A + (1-alpha) * B + beta * Reg)
54
55    # Init problem
56    prob = cp.Problem(obj, constraint)
57    prob.solve()    # Solving
58
59    return M.value
```

Listing 2: Layer-wise mask optimization

# D Layerwise Shuffling Experiments

With each setting, at each sparsity ratio, we seek subnetworks with 5 different seeds, and with each seed, we randomly shuffle the subnetwork two times. In addition to effective path ratios and effective node ratios, we compute the number of effective parameters after pruning and the actual remaining ones, then calculate the ratio between these two values.

Similar to [17] results, the performance and the number of effective parameters of high-density subnetworks after permuting the connections are similar to or even higher (in SNIP) than the unmodified ones. However, when the sparsity level becomes more intensive, the configuration of subnetworks is more concrete. Randomly rearranging connections within layers destroys this strict structure by detaching important edges, which drastically reduces the number of effective paths. The shuffled subnetworks lack input-output paths to transfer information during training, leading to a drop in performance compared with unmodified ones.

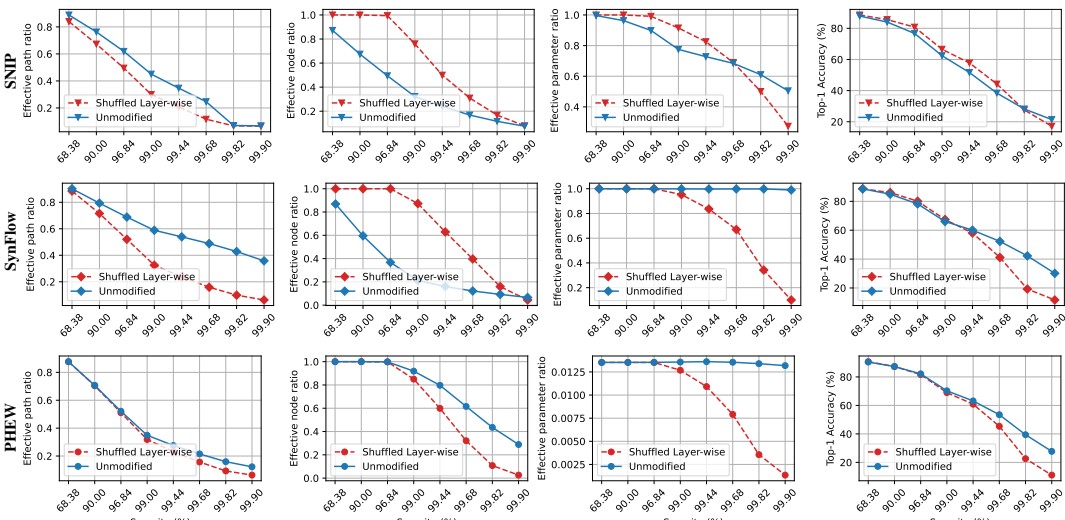

Figure 6: Layerwise shuffling results on various sparse subnetworks of ResNet20 produced SNIP, SynFlow, and PHEW at initialization on CIFAR-10.

# E  Pruning Time and FLOPs Reductions

Since NPB optimizes layer by layer, the complexity of NPB depends on network architecture (i.e., the number of layers and the size of each layer). With large layers, we sidestep the time-consuming issues by making a further step where we divide the nodes in the next layer into chunks with the same sparsity constraint, and optimizing chunks in a layer can be solved in parallel. Particularly, we consider layer $l$ with mask $\mathbf{m}^{(l)} \in \mathbb{R}^{h^{(l)} \times h^{(l+1)}}$ in which $h^{(l)}$ and $h^{(l+1)}$ are the number of nodes in layer $l$ and $l + 1$. We divide $h^{(l+1)}$ nodes into $K$ equal chunks. Instead of directly solving for $\mathbf{m}^{(l)}$, we solve $K$ problems $[\mathbf{m}_1^{(l)}, \mathbf{m}_2^{(l)}, ..., \mathbf{m}_K^{(l)}]$ where $\mathbf{m}_k^{(l)} \in \mathbb{R}^{h^{(l)} \times h_k^{(l+1)}}$. Thank for available convex optimization libraries we can find subnetworks more efficiently and quickly. We have computed the pruning time of our proposed method and compared it with other PaI methods in Tables 3, 4, and 5.

Besides pruning time, we find that the FLOPs reduction of subnetwork after pruning is more important in the context of pruning before training. We have measured FLOPs of subnetworks produced by different methods in Tables 3, 4, and 5. The result indicates that our NPB can produce subnetworks with lower FLOPs than other baselines while NPB outperforms PaI methods.

Table 3: Accuracy, pruning time (in seconds) and FLOPs of subnetworks for different pruning methods and compression ratios on Resnet18 - Tiny-ImageNet.

| | Accuracy (%) | | | | Pruning time (seconds) | | | | FLOPs ($10^8$) | | | |
|---|---|---|---|---|---|---|---|---|---|---|---|---|
| Sparsity (%) | 68.38 | 90.00 | 96.84 | 99.00 | 68.38 | 90.00 | 96.84 | 99.00 | 68.38 | 90.00 | 96.84 | 99.00 |
| SNIP | 56.99 | 53.43 | 48.77 | 36.02 | 5.14 | 4.95 | 5.55 | 5.64 | 11.35 | 5.77 | 3.04 | 1.55 |
| Iter-SNIP | 56.73 | 53.60 | 48.55 | 36.42 | 229.16 | 235.34 | 233.19 | 231.23 | 10.73 | 7.05 | 3.98 | 1.97 |
| SynFlow | 56.71 | 54.68 | 49.03 | 39.79 | 108.17 | 96.18 | 91.15 | 92.60 | 14.71 | 8.91 | 4.24 | 1.50 |
| PHEW | 58.09 | 55.93 | 50.81 | 40.54 | 5511.20 | 1342.03 | 471.23 | 324.78 | 14.29 | 8.35 | 3.92 | 1.50 |
| NPB | **58.39** | **56.82** | **51.37** | **41.05** | 380.52 | 375.65 | 384.32 | 387.89 | 14.37 | 5.21 | 1.74 | 0.59 |

Table 4: Accuracy, pruning time (in seconds) and FLOPs of subnetworks for different pruning methods and compression ratios on VGG19 - CIFAR-100.

| | Accuracy (%) | | | | Pruning time (seconds) | | | | FLOPs ($10^7$) | | | |
|---|---|---|---|---|---|---|---|---|---|---|---|---|
| Sparsity (%) | 68.38 | 90.00 | 96.84 | 99.00 | 68.38 | 90.00 | 96.84 | 99.00 | 68.38 | 90.00 | 96.84 | 99.00 |
| SNIP | 71.31 | 70.65 | 67.82 | 61.77 | 5.15 | 4.96 | 5.12 | 4.55 | 17.952 | 7.806 | 3.686 | 1.816 |
| Iter-SNIP | 72.84 | 70.86 | 67.72 | 63.13 | 115.91 | 115.52 | 116.83 | 117.60 | 18.465 | 9.479 | 4.951 | 2.529 |
| SynFlow | 71.63 | 69.18 | 66.98 | 62.11 | 96.55 | 100.33 | 101.90 | 104.67 | 22.998 | 12.702 | 6.306 | 2.605 |
| PHEW | 73.18 | 70.70 | 68.18 | 64.43 | 6928.59 | 1699.80 | 605.65 | 417.25 | 22.108 | 11.746 | 5.611 | 2.340 |
| NPB | **74.05** | **71.76** | **68.87** | **64.82** | 430.52 | 438.20 | 412.16 | 425.33 | 22.035 | 8.773 | 2.874 | 1.046 |

Table 5: Accuracy, pruning time (in seconds) and FLOPs of subnetworks for different pruning methods and compression ratios on Resnet20 - CIFAR-10.

| | Accuracy (%) | | | | Pruning time (seconds) | | | | FLOPs ($10^6$) | | | |
|---|---|---|---|---|---|---|---|---|---|---|---|---|
| Sparsity (%) | 68.38 | 90.00 | 96.84 | 99.00 | 68.38 | 90.00 | 96.84 | 99.00 | 68.38 | 90.00 | 96.84 | 99.00 |
| SNIP | 87.88 | 84.02 | 76,72 | 62.03 | 1.42 | 1.40 | 1.87 | 1.68 | 17,952 | 8.323 | 3.470 | 1.709 |
| Iter-SNIP | 88.17 | 84.22 | 77.05 | 64.95 | 58.21 | 54.60 | 52.02 | 57.61 | 18,465 | 9.698 | 4.510 | 2.022 |
| SynFlow | 88.64 | 84.94 | 78.22 | 66.05 | 57.46 | 56.24 | 53.43 | 54.13 | 22,998 | 11.549 | 4.263 | 1.633 |
| PHEW | 90.38 | 87.41 | **81.05** | 70.44 | 78.31 | 18.09 | 4.78 | 2.58 | 22,108 | 10.690 | 4.110 | 1.640 |
| NPB | **90.69** | **87.61** | 80.55 | **70.70** | 20.10 | 23.91 | 21.51 | 21.13 | 22,035 | 7.642 | 2.645 | 1.122 |

# F   Additional Results with PaI comparison

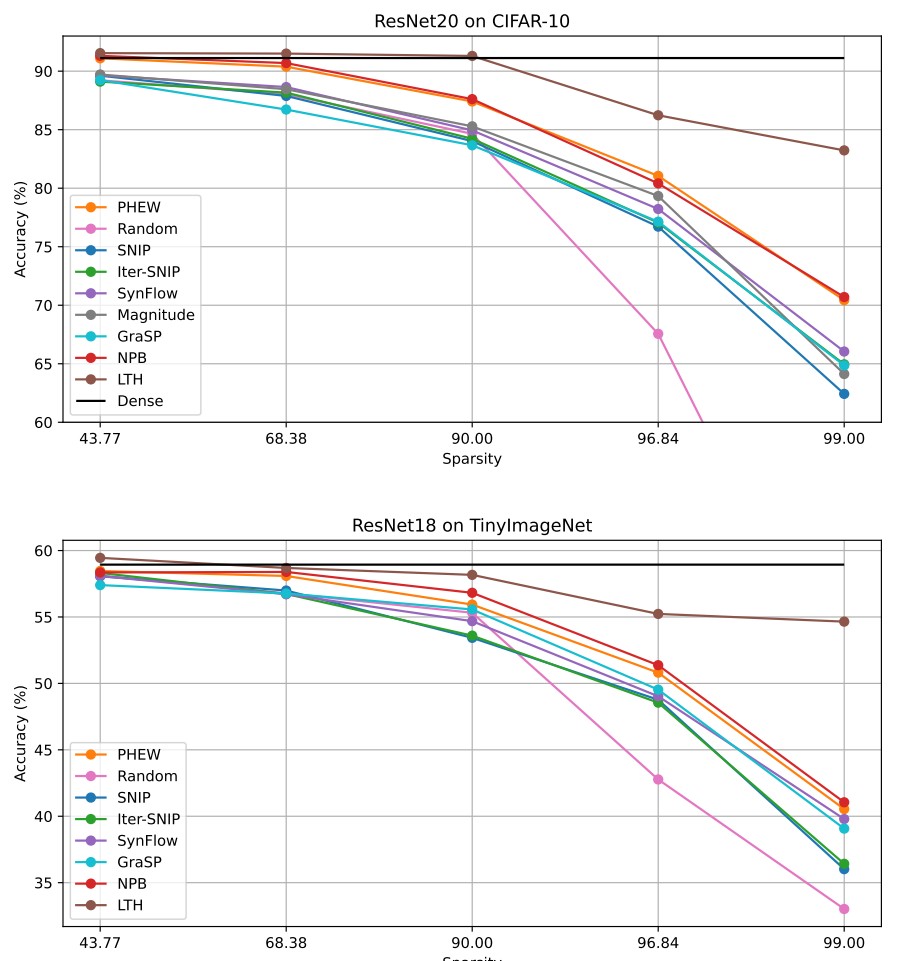

Figure 7: Accuracy of subnetworks found by different PaI methods and LTH at various sparsity levels on two settings ResNet20 on CIFAR-10 and ResNet18 on TinyImagenet.

# G    Additional Results with Ablation Studies

In this section, we conduct experiments with different balancing coefficients (alpha) and regularization coefficients (beta) in setting ResNet18 on Tiny-Imagenet and visualize results in Figure 8. We run NPB with alpha in {0.01, 0.05, 0.1, 0.5, 0.75, 0.9} and beta in {0.1, 0.5, 0.75, 1.0, 1.5, 2.0}. In the first row, we visualize the performance of subnetworks generated by different alpha beta hyperparameters. In addition, to illustrate the correlation between subnetworks of NPB and PaI methods in terms of the number effective nodes and paths, we visualize alpha and beta subnetwork variants of NPB and PaI methods in the second row of Figure 8. Although NPB's variants have difference performance, they still outperform baselines. When visualize NPB variants with different PaI methods (SynFlow, Random, and PHEW) together, we can see that NPB variants concentrate around a specific region near PHEW which we hypothesize as a balancing area. Besides, networks in this region show a better results compared with SynFlow or Random, and have competitive performance with each others. This strengthens our Node-Path Balancing Principle.

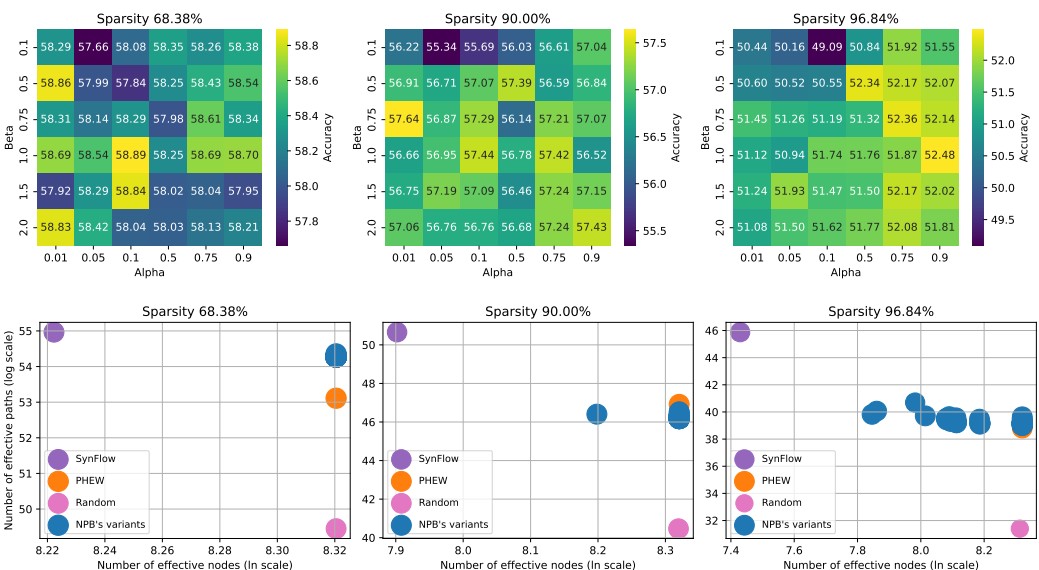

Figure 8: Ablation study on alpha and beta hyperparameter with setting ResNet18 on Tiny-Imagenet

NPB depends on the layer-wise sparsity ratios, disparate ratios guide the pruner to a diverse set of solutions. Then, it is essential to investigate the effect of sparsity initializers on NBP's performance. We follow Liu et al. [29] to use ERK [14], Uniform, Magnitude ratio, SNIP ratio found by applying SNIP [26], and GraSP ratio by using GraSP [45] as five methods to produces layer-wise sparsity levels. Figure 9 illustrates results with these sparsity initializers for ResNet18 on Tiny-Imagenet with different sparsity levels. With the same value of $\alpha$ and $\beta$, we observe that different layer-wise sparsifiers produce different network structures and performances as well. This difference raises a question for further work how we can specify ratios that help obtaining promising solutions.

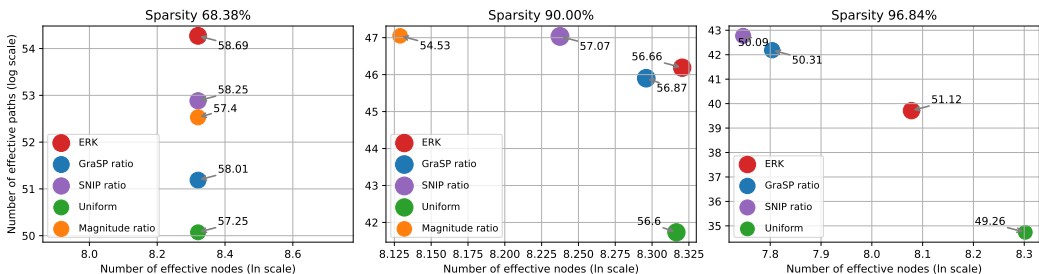

Figure 9: Ablation study on different layerwise sparsity initializer with setting ResNet18 on Tiny-Imagenet

# H  NAS-Bench-Macro Experiments

NAS-Bench-Macro [41] is a macro benchmark which is used in Neural Architecture Search. It includes 8 searching layers, each having three candidate blocks which construct 6,561 different networks with the parameter varying from 300k to 3M. The network structure is presented in Table 6. The structure is inspired from MobileNetV2 search space. In particular, candidate blocks are (i) Identity connection, (ii) MB3_K3 which is MobileNetV2 block with kernel size 3 and expansion ratio 3, and (iii) MB6_K5 which is MobileNetV2 block with kernel size 5 and expansion ratio 6. To make a fair comparison between candidates, we only compare networks with a similar number of parameters (i.e., in a range of 300k parameters). Note that, each candidate here is a dense network. We compute metrics in four different parameter ranges and visualize them in Figure 10.

Table 6: Macro structure of search space on NAS-Bench-Macro.

| n | input | block | channel | stride |
|---|---|---|---|---|
| 1 | $32 \times 32 \times 3$ | $3 \times 3$ conv | 32 | 1 |
| 2 | $32 \times 32 \times 32$ | Choice Block | 64 | 2 |
| 3 | $16 \times 16 \times 64$ | Choice Block | 128 | 2 |
| 3 | $8 \times 8 \times 128$ | Choice Block | 256 | 2 |
| 1 | $4 \times 4 \times 256$ | $1 \times 1$ conv | 1280 | 1 |
| 1 | $4 \times 4 \times 1280$ | global avgpool | – | – |
| 1 | 1280 | FC | 10 | – |

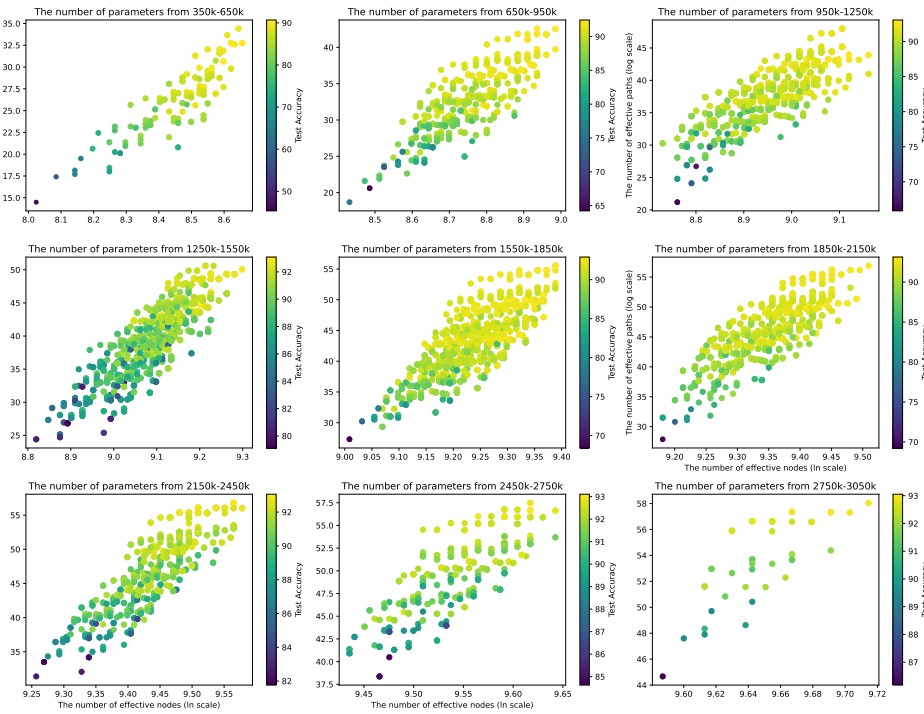

Figure 10: The accuracy of network candidates in NAS-Bench-Macro benchmark along with the number of activated nodes and paths in different parameter ranges.

