# OpenReview forum: "Towards Data-Agnostic Pruning At Initialization: What Makes a Good Sparse Mask?"
_NeurIPS.cc/2023/Conference — NeurIPS 2023 poster_

### Official Review · Reviewer_ygEG · 2023-07-03

**Soundness:** 3 good
**Presentation:** 2 fair
**Contribution:** 2 fair
**Rating:** 6
**Confidence:** 4

**Summary:**

The authors perform a large-scale empirical analysis of models from the NAS-Bench-Macro benchmark which motivates to the development of (1) the node-path balancing principle and (2) Node-Path Balancing Pruner (NPB) -- a data-agnostic pruning-at-initialization (PAI) scheme. At a high level, the node-path balancing principle suggests that networks that have a good balance between the number of nodes and edges on a continuous/unbroken path from the neural network input to the neural network have a higher capacity for performance (i.e. classification accuracy). Their pruning algorithm, NPB, then solves a constrained optimization problem to strike this node-path balance for a given sparsity level. The driving factor for this research is to design advanced PAI algorithms for producing efficient models with better performance (i.e., classification accuracy) than existing PAI schemes.

**Main contributions:**

* Node-path balancing principle: Empirical analysis of NAS-Bench-Macro benchmark yields new perspective and principle for guiding design of pruning-at-initialization (PAI) schemes.
* NPB: A novel data-agnostic PAI scheme designed using the node-path balancing principle which outperforms existing PAI methods at lower FLOPs during inference.

**Post-rebuttal revision:**

Following the rebuttal, I feel that the authors have clarified my questions and concerns. Accordingly, I am increasing my rating to weak accept with the expectation that they will accommodate my requested minor revisions.


**Strengths:**

* Large scale analysis of NAS-Bench-Macro benchmark models suggesting connection between model performance and balance of effective nodes and paths.
* **NPB pruning-at-initialization algorithm**: Produces results comparable to state-of-the-art PAI scheme PHEW with fewer FLOPs during inference.


**Weaknesses:**

* **Looseness/inexactness of the node-path balancing principle**: I found the statement/rigor of this guiding principle (which was used to inform the design of NPB) to be very inexact. The motivating plots in Figure 4 suggest that there is nuance between this node-path balance and the resulting classification accuracy, but this was not further explored. Specifically, some networks with similar node-path balance in Figure 4 appear to have large variations in classification accuracy.
* **Lack of theory for node-path balancing principle**: Developing theory to study the principle would help design/inform a more exact guiding principle for PAI schemes.
* **Figure clarity**: It was not always evident what the takeaway is from certain figures and I feel that the presentation of certain figures could be improved. I found myself staring at some figures for a while trying to decipher what was important/meaningful as there is a lot of data. This isn’t to say there is not meaningful information in the figures but I feel like more careful curation/design of the figures could be enacted. For suggestions/specific figures, see comments under “Questions”.


**Questions:**

1. How did you compute/measure the FLOPs presented in Table 1? I may have missed something, but I did not see an explanation in the main body or in the appendix.
2. Did you observe training/inference speedup due to reduced FLOPs? If so, it would be worth mentioning as NPB tends to require fewer FLOPs than other methods (at least at sparsity >= 90%).
3. In some instances, models appear to have very different balance of nodes and paths but comparable accuracy (e.g., ResNet20 on CIFAR-10 sparsity 99%, VGG19 on CIFAR-100 sparsity 90%) or similar balance of nodes and paths but very different accuracy (VGG19 on CIFAR-100 sparsity 68.38%, ResNet18 on Tiny-Imagenet sparsity 90%). How would your principle in its current form explain this?

Comments about figures:

* **Figure 3**: I had to stare at this for a while to figure out what the takeaway was. It seems that more effective nodes can help improve accuracy, but it is not enough on its own. I’m not sure if there is a different way to present the data to make it more readily apparent (and I’m not counting it against you) but I just wanted to mention it.
* **Figure 4**: I think the scale of the accuracy colormap should be the same across all 4 plots (i.e. darkest blue is min accuracy across all 4 plots ~65% and yellow is max across all 4 plots ~93%). Then accuracy diversity across the four groups would be more apparent (i.e. more params -> higher accuracy) and you could just have a single colorbar.
* **Figure 5**: For each dataset (i.e. row) I think the x- and y-axis range should be the same across the four plots. This revision should better highlight trends/changes in node-path balance with increasing sparsity.


**Limitations:**

Repeating a weakness here as I believe it is a limitation that is not acknowledged/discussed:

* **Looseness/inexactness of the node-path balancing principle**: I found the statement/rigor of this guiding principle (which was used to inform the design of NPB) to be very inexact. The motivating plots in Figure 4 suggest that there is nuance between this node-path balance and the resulting classification accuracy, but this was not further explored. Specifically, some networks with similar node-path balance in Figure 4 appear to have large variations in classification accuracy.

---

> ### Author Rebuttal · Authors · 2023-08-09
>
> We would like to thank you for the time to review our work. We would like to address all weaknesses pointed out by you point-by-point below:
> > Q1. Looseness/inexactness of the node-path balancing principle
> >
>
> A1: In Neural Architecture Search (NAS) context, it's important to consider that other aspects beyond node-path balance can contribute to final classification accuracy. In the case of NAS-Bench-Macro, networks with similar node-path balance but varying classification accuracies can be affected by other architectural configurations, such as differences in pooling layers, kernel sizes, and expansion ratios in the MobileNet-v2 block. These architectural variances often result in different numbers of parameters, influencing the overall network performance.
>
> However, in pruning context, pruning methods focus on maintaining the same network structure while pruning connections within the network based on specific sparsities. Consequently, our experiments did not explore other architectural elements beyond node and path balance.
>
> We emphasize that our experiments with NAS demonstrated that networks with balanced nodes and paths tend to exhibit superior performance. Based on these findings, we have taken the step to use the balance between two metrics as a fundamental criterion for designing pruning methods.
>
> > Q2. Lack of theory for node-path balancing principle.
> >
>
> A2: We agree that our paper is not a theoretical study. However, the underlying motivations for our method are justified and easy to understand. Let us recall that focusing on optimization of either nodes or paths will lead to a loss on the remaining quantity, which leads to suboptimal performance. This statement is intuitive and we think that unnecessary theoretical analysis would add little to our paper at this stage. The key theoretical challenge here is the problem of identifying the balancing points and the precise relation between node-path and performance. These are difficult research problems in their own rights, and we postpone them to future work.
>
>
> > Q3. How did you compute/measure the FLOPs presented in Table 1?
> >
>
> A3: We appreciate your inquiry regarding the computation of FLOPs presented in Table 1. To derive the FLOPs values, we adopted the methodology outlined in [1]. It's worth noting that the FLOPs calculation is performed on a layer-by-layer basis. At each layer, the FLOPs are computed by considering the number of parameters, the input size of the layer, and the layer's sparsity.
> We will incorporate the explanation of the FLOPs measurement methodology in the revised version of the paper.
>
> [1] https://github.com/Eric-mingjie/rethinking-network-pruning/blob/master/cifar/weight-level/count_flops.py
>
> > Q4. Did you observe training/inference speedup due to reduced FLOPs? If so, it would be worth mentioning as NPB tends to require fewer FLOPs than other methods (at least at sparsity >= 90%).
> >
>
> A4: We appreciate your interest in the potential speedup resulting from reduced FLOPs in our proposed method. In the case of NPB, which employs unstructured pruning similar to other PaI methods, we currently provide theoretical FLOPs measurements rather than real-world speedup figures. This limitation arises from the constrained support available from off-the-shelf GPUs for unstructured pruning techniques.
>
> However, it's important to note that while unstructured pruning may not exhibit real-time speedups presently, it remains a valuable mathematical framework and an empirical testbed for exploring novel Sparse Neural Network algorithms. Furthermore, there is a growing trend of improved practical support for unstructured pruning in the field [2].
>
> [2] https://arxiv.org/abs/2302.02596
>
> > Q5. In some instances, models appear to have very different balance of nodes and paths but comparable accuracy (e.g., ResNet20 on CIFAR-10 sparsity 99%, VGG19 on CIFAR-100 sparsity 90%) or similar balance of nodes and paths but very different accuracy (VGG19 on CIFAR-100 sparsity 68.38%, ResNet18 on Tiny-Imagenet sparsity 90%). How would your principle in its current form explain this?
> >
>
> A5: The disparities in accuracy, as you noted, can stem from various factors such as the number of effective parameters (as defined in Appendix B), the distribution of unpruned weights within kernels, and more. These intricacies can lead to differences in performance despite comparable or dissimilar node-path balances.
>
> However, a general trend can still be discerned. We find that when the number of active neurons and input-output paths of a sparse network align within the balancing range, there tends to be a likelihood of achieving superior performance after training.
>
> > Q6. Figure 3: I had to stare at this for a while to figure out what the takeaway was. It seems that more effective nodes can help improve accuracy, but it is not enough on its own.
> >
>
> A6: Thank you for your time and consideration. Indeed, Figure 3 shows more nodes can help enhance performance but it is not always the case. Apart from nodes, paths play an important role in the quality of subnetworks. These two metrics mutual impact on each other. Therefore, we conjecture that to create optimal subnetwork, we should take both nodes and paths into account.
>
> > Q7. Figure 4: I think the scale of the accuracy colormap should be the same across all 4 plots. Then accuracy diversity across the four groups would be more apparent (i.e. more params -> higher accuracy) and you could just have a single colorbar.
> >
>
> A7: Thank you for your suggestions to improve our paper. We will revise the figure and update it in the next version.
>
> > Q8. Figure 5: For each dataset (i.e. row) I think the x- and y-axis range should be the same across the four plots. This revision should better highlight trends/changes in node-path balance with increasing sparsity.
> >
>
> A8: Thank you for your suggestions to improve our paper. We will revise the figure and update it in the next version.

---

> > ### Comment · Reviewer_ygEG · 2023-08-11
> > **Response to authors**
> >
> > I would like to thank the authors for their responses to my review. I respond inline below.
> >
> > > A1: In Neural Architecture Search ...
> >
> > Thank you for clarifying. This insight resolves my concern with Figure 4 (i.e., the varying architectural configurations in NAS-Bench-Macro can account for variations in accuracy for similar node-path balance). For clarity, I recommend including a sentence noting this point somewhere in the final version of the paper (if there is not already a discussion on this in the paper that I missed).
> >
> > > A2: We agree that our ...
> >
> > Thanks for addressing this. I agree with your response and acknowledge that it is perhaps beyond the scope of this paper to establish theory for the relationship between node-path balance and performance.
> >
> > > A3: We appreciate your inquiry ...
> >
> > Thanks for the explanation and reference. I agree that this should be noted in the revised version of the paper.
> >
> > > A4: We appreciate your interest in ...
> >
> > Thank you for your clarification. I did not mean this comment to detract from your contribution but, rather, see if you could highlight it as a benefit of your approach. I agree that unstructured sparsity has its merits (as highlighted in question 3.5 of your reference).
> >
> > > A5: The disparities in accuracy, as you noted, can stem from various factors such as the number of effective parameters (as defined in Appendix B), the distribution of unpruned weights within kernels, and more. These intricacies can lead to differences in performance despite comparable or dissimilar node-path balances.
> > However, a general trend can still be discerned. We find that when the number of active neurons and input-output paths of a sparse network align within the balancing range, there tends to be a likelihood of achieving superior performance after training.
> >
> > Thank you for your response. I understand and acknowledge that many factors can play into the resulting accuracy. I recommend mentioning this somewhere (perhaps the discussion around Figure 5 or in the conclusion) as a limitation of the node-path balancing principle because, in your own words, “These intricacies can lead to differences in performance despite comparable or dissimilar node-path balances”. You could also mention that future/follow-up research could seek to better understand this nuance.
> >
> >
> > > A7&A8: Thank you for your suggestions to improve our paper. We will revise the figure and update it in the next version.
> >
> > Thanks.
> >
> > I feel that the authors have clarified my questions and concerns. Accordingly, I am increasing my rating to weak accept with the expectation that they will accommodate my requested minor revisions.

---

> > > ### Author Response · Authors · 2023-08-11
> > > **Response to Reviewer ygEG**
> > >
> > > We sincerely appreciate your quick response and invaluable feedback. We will definitely incorporate your suggestions into the next version of the paper.

---

### Official Review · Reviewer_HuF1 · 2023-07-07

**Soundness:** 3 good
**Presentation:** 3 good
**Contribution:** 3 good
**Rating:** 6
**Confidence:** 4

**Summary:**

This paper examines Pruning at Initialization (PaI) methods using two novel metrics: the number of effective paths and the number of effective nodes. The authors find that layer reshuffling negatively impacts the performance of sparse neural networks obtained through PaI methods in the extreme sparsity regime. Based on the previous finding, authors present a novel data-agnostic PaI method, Node-Path Balancing Principle (NPB), which achieves the SOTA performance by effectively balancing the two proposed metrics.

**Strengths:**


- This paper provides a clear rationale for introducing the two proposed metrics; the authors empirically demonstrate the importance of the metrics via two experiments. First, experiments conducted on the NAS benchmark reveal a strong correlation between the two metrics and the final performance of sparse networks. Secondly, through layer shuffling experiments, the authors illustrate that simply increasing the number of effective nodes in the extreme sparsity regime is insufficient to prevent performance degradation due to the sharp decline in the number of effective paths.
- The proposed method demonstrates superior performance compared to the baselines, and the paper covers a fair amount of relevant previous studies.
- The paper is effectively structured and exhibits clear and concise writing


**Weaknesses:**

- While many previous PaI methods prioritize the weight magnitude as the importance metric, the main motivation behind NPB focuses on the topology of the sparse network. Thus, the ablation study with respect to different weight initialization would strengthen the paper. Further, is NPB robust to *weight reinitialization*?
- While the layer shuffling experiments are well-justified, it is confined to a single configuration (CIFAR-10, ResNet20). I wonder if similar observations can be made under different experimental settings. Also, does layer-wise shuffled NPB exhibit similar behavior as to layer-wise shuffled PHEW?
- There are two points in need of clarification. First, in lines 332-333, the term "chunks" is unclear, and it is not apparent how the parallel computation can be effectively achieved. Second, in the case of convolutional layers mentioned in lines 261-263, it is unclear whether equations 3-5 are still applicable when the mask vector is not boolean.
- Although the authors claim that NPB generally outperforms PHEW, the standard deviation with respect to random seeds should be included in the paper for fair comparison. For instance, the performance gain depicted in Fig. 7 seems marginal. Additionally, in lines 694-695, it is stated that NPB consistently outperforms the baselines regardless of the hyperparameter configuration. However, it should be noted that certain choices may lead to worse performance compared to the baselines (refer to Table 1).

I am willing to raise my score if above concerns are well addressed during rebuttal.


**Questions:**

- In equation 3, what is the purpose of “min” operation? Or is it just a typo of $\min(m_{ij}^l - 1, 0)$?
- What kind of optimizer is implemented in the code as to approximating the original integer programming formulation?

**Limitations:**

The paper presents a future research direction in lines 706-707, which is included in the Appendix. However, it is equally important for the authors to acknowledge and discuss the limitations of their proposed method.

---

> ### Author Rebuttal · Authors · 2023-08-09
>
> We would like to thank you for the time to review our work. We would like to address all weaknesses pointed out by you point-by-point below:
> > Q1. While many previous PaI methods prioritize the weight magnitude as the importance metric, the main motivation behind NPB focuses on the topology of the sparse network. Thus, the ablation study with respect to different weight initialization would strengthen the paper. Further, is NPB robust to weight reinitialization?
> >
> A1: We appreciate your suggestion for enhancing our paper. In response, we conducted experiments with different weight initializations for NPB on setting ResNet20 on CIFAR-10 and visualized it in Figure 2 in attached file. In our paper, we use Kaiming Normal initialize network weight. The result shows that different weight initializations only affect the performance of subnetworks. While the number of effective nodes remains relatively consistent across different initializations, a minor variation can be observed in the number of effective paths between subnetworks. This discrepancy is attributed to the distribution of remaining weights within the convolutional layers' kernels. This experiment demonstrates that NPB exhibits robustness to different weight reinitializations.
> > Q2. While the layer shuffling experiments are well-justified, it is confined to a single configuration (CIFAR-10, ResNet20). I wonder if similar observations can be made under different experimental settings.
> >
> A2: We have run additional layer-wise shuffling experiments with VGG19 on CIFAR-10 with SynFlow and SNIP. We visualize the results on Figure 3 in the attached file. In general, this experiment shows the same behavior with which in setting ResNet20 on CIFAR-10.
> > Q3. Also, does layer-wise shuffled NPB exhibit similar behavior as to layer-wise shuffled PHEW?
> >
> A3:  We have run shuffling experiments for NPB with setting ResNet20 on CIFAR-10 and reported in Table 1 in attached file. After shuffling subnetworks found by NPB, the subnetworks' width broaden while the figure for paths drops significantly as the level of sparsity increases. However, similar to PHEW, the performance of unmodified subnetworks is better than the shuffled counterparts.
> > Q4. There are two points in need of clarification. First, in lines 332-333, the term "chunks" is unclear, and it is not apparent how the parallel computation can be effectively achieved. Second, in the case of convolutional layers mentioned in lines 261-263, it is unclear whether equations 3-5 are still applicable when the mask vector is not boolean.
> >
> A4: To be more specific, we consider layer $l$ with mask $\mathbf{m}^{(l)} \in \mathbb{R}^{h^{(l)} \times h^{(l+1)}}$ in which $h^{(l)}$ and $h^{(l+1)}$ are the number of nodes in layer $l$ and $l+1$. We divide $h^{(l+1)}$ nodes into $K$ equal chunks.
> Instead of directly solving for $\mathbf{m}^{(l)}$, we solve $K$ problems $[\mathbf{m}^{(l)}_1, \mathbf{m}^{(l)}_2,  ..., \mathbf{m}^{(l)}_K ]$ where $ \mathbf{m}^{(l)}_k \in \mathbb{R}^{h^{(l)} \times h^{(l+1)}_k}$.
>
> With convolutional layers, we consider a kernel as a connection in linear layer (please refer to Figure 1). The mask $\mathbf{M} \in \mathbb{R}^{c_{in}, c_{out}, h, w}$ with binary entries will transform to $\mathbf{M}’ \in \mathbb{R}^{c_{in}, c_{out}}$ with an entry $m_{i,j} \in \mathbb{N}$ with values from 0 to hw. Then, we treat M as an integer variable, and we leverage an available optimization library (CVXPY library) to solve the mixed integer programming problem.
> > Q5. Although the authors claim that NPB generally outperforms PHEW, the standard deviation with respect to random seeds should be included in the paper for fair comparison. For instance, the performance gain depicted in Fig. 7 seems marginal. Additionally, in lines 694-695, it is stated that NPB consistently outperforms the baselines regardless of the hyperparameter configuration. However, it should be noted that certain choices may lead to worse performance compared to the baselines (refer to Table 1).
> >
> A5: We appreciate your feedback. We will incorporate standard deviation in the next version. Regarding the statement in lines 694-695, we appreciate your observation. We intend to clarify our claim to accurately reflect that NPB generally outperforms the baselines, and while certain hyperparameter configurations might lead to slightly reduced performance compared to PHEW. Overall, we believe that our findings still highlight the competitiveness and effectiveness of NPB.
> > Q6. In equation 3, what is the purpose of “min” operation? Or is it just a typo of $min(m_{i,j}^l - 1, 0)$
> >
> A6: The "min" operation in Eq. 3 serves the purpose of transforming the formulation to a version that is compatible with convex optimization library. We acknowledge the error, it indeed should be $min(m_{i,j}^l - 1, 0)$. We will rectify it in the next version.
> > Q7. What kind of optimizer is implemented in the code as to approximating the original integer programming formulation?
> >
> A7: The optimization problem is solved efficiently via the available convex optimization library (CVXPY library) in which we use the default solver for mixed integer programming named SCIPY [1].
>
> [1] https://www.cvxpy.org/tutorial/advanced/index.html#setting-solver-options
> > Q8. It is equally important for the authors to acknowledge and discuss the limitations of their proposed method.
> >
> A8: We appreciate your feedback. We would like to acknowledge the following limitations of our approach:
> - Global Optimization of Node-Path Balancing: Our method has not achieved global optimization of node-path balancing. The intricate relationship between effective nodes and paths presents a complex optimization challenge that remains a subject of ongoing exploration.
> - Discrete Optimization: The current form of our method involves discrete optimization. This can potentially limit applications such as applying NPB as a criterion in prune during training, or in Neural Architecture Search problems.

---

> > ### Comment · Reviewer_HuF1 · 2023-08-14
> >
> > I appreciate the authors for their detailed response and the supplementary experiments to confirm the effectiveness of NPB. That said, I am now confused with the author's following response regarding Table 1 in the pdf.
> >
> > > A3: We have run shuffling experiments for NPB with setting ResNet20 on CIFAR-10 and reported in Table 1 in attached file. After shuffling subnetworks found by NPB, the subnetworks' width broaden while the figure for paths drops significantly as the level of sparsity increases. However, similar to PHEW, the performance of unmodified subnetworks is better than the shuffled counterparts.
> >
> > I believe layer-wise shuffling experiment is a critical motivating experiment behind NPB along with NAS experiment. To my understanding, the shuffled versions of SNIP and SynFlow in Figure 3 (main paper): (1) occasionally outperforms original one due to increased number of effective nodes under 99% sparsity - *importance of effective nodes*; (ii) hurts performance due to insufficient number of effective paths at extreme sparsities (i.e., > 99%) - *importance of effective paths*. However, Table 1 (in pdf) shows shuffling NPB leads to an increase in the number of effective nodes and a decrease in the number of effective paths, while I expected reduction in both measurements. Further, authors point out the performance of unshuffled network is better than the shuffled counterparts, but I do not find this explanation satisfactory because layer-shuffling consistently hurts performance of PHEW, and SNIP/SynFlow at extreme sparsities as well. From my perspective, Table 1 appears to emphasizes the significance of effective paths alone rather than achieving a *balance* between nodes and paths. Additionally, is there any particular reason as to presenting the NPB results in terms of number of effective nodes and paths rather than the previously employed ratio?

---

> > > ### Author Response · Authors · 2023-08-14
> > > **Response to Reviewer HuF1**
> > >
> > > Many thanks for your further comments. We are sorry that our response still contains confusions. We hope the following explanation will clarify these confusions:
> > >
> > > Regarding results presented in Table 1 (extra pdf):
> > >
> > > 1. Note that a possible explanation to why shuffling after NPB produces those results in Table 1 (extra pdf) is that the optimization method used in NBP eventually reaches a *local optimum for node-path balancing*. That is, applying shuffling after our NPB means that we are moving away from that local optimum. In particular, while shuffling may improve the representation capacity (by increasing the number of effective nodes), it substantially decreases number of paths compared to that of NPB, as seen in that table, which limits the information flow. As a result, the performance of shuffled subnetwork drops (as seen in sparsity levels ~96% and 99%). Therefore, Table 1 (extra pdf) is just an evidence that shuffling moves us away from that local optimum, which eventually also results in a performance decrease.
> > >
> > > 2. We would like to highlight that the main claim of our paper is that NPB is well calibrated to achieve the effective node-path balance. As discussed in Point 1, Table 1 (extra pdf) indicates that layer-wise shuffling may have negative impact on such balance, leading to deteriorated performance. Please note that here we argue that it is the node-path balance that matters, rather than the absolute numbers of effective nodes and paths. Therefore, we observed layer-wise shuffling may either reduce the number of effective paths with the same number of effective nodes (non-extreme sparsity cases), or occasionally increase the number of effective nodes but decrease the number of effective paths (since shuffling uniformly redistributes edges to all node).
> > >
> > > 3. Now, you are correct that only based on the results in Table 1 (extra pdf) we may interpret that just increasing the number of effective paths may improve the performance. However, this conclusion is not correct in general, as in the main paper we have already showed that increasing the number of effective paths alone is not enough (please refer to Figure 5 in our paper): For instance, subnetworks generated by SynFlow have much higher number of effective paths and fewer effective nodes, but with lower accuracy, compared to NPB. Therefore, results in Table 1 should not be interpreted alone, which actually complements our main findings in the paper and further support the argument that NPB is better calibrated to achieving the desired node-path balance.
> > >
> > > 4. Overall, we should not read Table 1’s results alone and draw final conclusions from it. Those results are just auxiliary ones to further support our claim that NPB is better calibrated to achieving the desired balance.
> > >
> > >
> > >
> > > Regarding using numbers of effective nodes and paths instead of ratio:
> > >
> > > Many thanks for this comment. Here we used the numbers rather than ratio just to better visualise the differences between competing methods. We can provide both versions (number and ratio) in appendix for the sake of consistency.

---

> > > > ### Comment · Reviewer_HuF1 · 2023-08-14
> > > >
> > > > Thank you for the prompt reply. I carefully read your response, but my confusion regarding layer-shuffling experiment still remains.
> > > >
> > > > - Both PHEW and NPB shows good performance in Figure 5 (main paper). The difference is that layerwise-shuffled PHEW leads to reduction in both effective nodes and paths while NPB follows the same trend as that of SNIP and SynFlow (re. Table 1 in pdf & Figure 3 in paper).
> > > >
> > > > Regarding above obervation, what leads to say that NPB have achieved better balance than PHEW? Is it solely based on NPB's higher final accuracy? Please correct me for any misunderstanding.

---

> > > > > ### Author Response · Authors · 2023-08-15
> > > > > **Response to Reviewer HuF1**
> > > > >
> > > > > Yes, we believe there is a misunderstanding in this case: We did not claim that NPB achieves better balancing than PHEW. We only showed that they both achieve good balancing, hence their good performance.
> > > > >
> > > > > Your observation from the figures is correct, but that in fact shows the differences in the working mechanisms of PHEW and NPB, and it does not imply that NPB finds better balancing points. In particular, here’s a more detailed clarification:
> > > > >
> > > > > *An explanation for your observation about layer-shuffling experiments*:
> > > > >
> > > > > Your observation: "The difference is that layer-wise shuffled PHEW leads to reduction in both effective nodes and paths while NPB follows the same trend as that of SNIP and SynFlow (re. Table 1 in pdf & Figure 3 in paper)." is correct, the explanation for this is the difference between working mechanisms of PHEW and NPB, which goes as follows:
> > > > >
> > > > > PHEW focuses on increasing the number of effective nodes by gradually adding new paths such that the network is as wide as possible. Consequently, reshuffling hurts the concrete network configuration and then reduces both the number of effective nodes and effective paths as sparsity is higher since shuffling creates disconnected paths (we mentioned in lines 188-190).
> > > > >
> > > > > On the other hand, the objective of our method NPB is to focus on balancing the number of nodes and paths. When sparsity level increases, due to the limited number of parameters, the number of activated nodes reduces. As a result, shuffling widens the subnetwork since it redistributes the edges to nodes.
> > > > >
> > > > > We notice that such observation is not the reason to claim that NPB achieves better balancing. It just explains that due to their differences, PHEW and NPB behave differently under the layer-shuffling experiments.
> > > > >
> > > > > To be more precise, notice that for sparsity levels lower than 96.84% both PHEW and NPB seems to find similar balancing points. However, for sparsity levels 96.84% and onwards, NPB achieves balancing points with lower effective nodes, compared to that of PHEW. This is the reason when we do layer-wise shuffling, the shuffled version of NPB has more number of nodes in than its unshuffled counterpart, while this number decreases for PHEW.
> > > > >
> > > > > *On region of node-path balancing points*:
> > > > >
> > > > > Both PHEW and NPB can find subnetworks with good node-path balancing, we do not claim that NPB is better than PHEW at this point, and our layer-shuffling experiments also do not imply such claim.
> > > > >
> > > > > In order to claim that one PaI method achieves better node-path balancing than another method, one needs to build a metric to measure the level of balancing. However, finding such metric or even finding optimal balancing points is challenging task, because it depends  on various factors such as architecture’s configuration, sparsity level, etc., and thus, remains an interesting future work.
> > > > >
> > > > > In our paper, we empirically demonstrated that there are specific regions of balance between effective nodes and paths, where subnetworks tend to perform better. Our proposed Node-Path Balancing principle is grounded in the observation that networks with a balance between these two metrics exhibit superior performance.
> > > > >
> > > > > While PHEW may achieve certain levels of balance leading to good performance, our focus on explicitly balancing nodes and paths allows NPB to consistently achieve better results across a range of experiments.
> > > > >
> > > > > Our method actively aims to strike better balancing between effective nodes and paths, allowing for improved performance beyond what might be achieved by approaches like PHEW, which do not directly optimize for this specific balance.

---

> > > > > > ### Comment · Reviewer_HuF1 · 2023-08-21
> > > > > >
> > > > > > I really appreciate your thorough response. I believe that the motivation regarding layer-shuffling experiment need to be greatly improved in the final manuscript. Of note, the reason as to why NPB shows better performance compared to PHEW should also be clearly explained.

---

> > > > > > > ### Author Response · Authors · 2023-08-21
> > > > > > > **Response to Reviewer HuF1**
> > > > > > >
> > > > > > > We sincerely appreciate your valuable feedback and discussion. We will certainly incorporate your suggestions into the next version of our paper.

---

### Official Review · Reviewer_qyia · 2023-07-10

**Soundness:** 3 good
**Presentation:** 4 excellent
**Contribution:** 3 good
**Rating:** 6
**Confidence:** 3

**Summary:**

This paper posits that the performance of (neural network) pruning at Initialization methods depends on a balance between effective nodes and paths. With this framework, authors explain why randomly shuffled subnetworks are sometimes more effective than subnetworks found by pruning at initialization methods. Finally, using the Node-Path balancing principle, the authors propose a pruning at initialization scheme by solving a linear program that optimizes pruning mask to balance effective path and nods, which outperforms other pruning at initialization methods.


Empirical experiments are done on image datasets (CIFAR and TinyImagenet) with VGG and Resnet-20 models. Further empirical evidence for the node-path balancing principle is demonstrated through NAS benchmarks.

**Strengths:**

- The paper is well-written, and generally content is well-organized.
- The proposed node-path balancing principle is used to explain an existing phenomenon (random shuffling) as well as present a new pruning at initialization scheme, which is quite interesting.
- The convex program-based pruning method (NPB) outperforms other pruning at initialization methods and requires fewer FLOPs. PHEW is close in performance to NPB. However, the node-path balancing framework can also explain that, is interesting.

Overall, the paper does a good job of introducing the node-path balancing principle and provides several empirical evidence in its support.

**Weaknesses:**


- The NAS experiments are interesting, but it wasn't clear to me to connect them with pruning at initialization or data-agnostic pruning, as NAS experiments are data-dependent.
- The term balance may be vaguely used.
   - The proposed principle states that at a particular sparsity, the best-performing subnetwork has to strike a balance between the effective nodes and paths. However, it does not predict the right balance.
    - It would be interesting to see if this balance depends on sparsity, model, and dataset or model and datasets only. Could the best-performing subnetwork network be maximizing average effective paths per node?
- The experiments are limited to image datasets and two neural network architectures. However, I appreciate the authors reporting results over three random seeds though.

**Questions:**

- Please see the weaknesses section.
- Shouldn't Eq 4/5 consider the mask M as binary? As such, M is a real value; how is the solution converted to a pruning mask?
- Line 262: Should M vary from 0 to hw instead of kw for convolution masks?


**Limitations:**

None, that are not discussed or addressed.

---

> ### Author Rebuttal · Authors · 2023-08-09
>
> We sincerely thank the reviewer for recognizing that our proposed method is technically sound and yielding commendable empirical results. We would like to address all the weaknesses pointed out by you point-by-point below:
>
> > Q1. The NAS experiments are interesting, but it wasn't clear to me to connect them with pruning at initialization or data-agnostic pruning, as NAS experiments are data-dependent.
> >
>
> A1: The intuition behind NAS experiments is that we would like to show that there is a strong correlation between the two metrics (node, path) and the final performance of network candidates. In particular, networks which are designed to have higher node and path tend to superior performance after training. We link this problem with sparse network configuration design.
>
> > Q2. The proposed principle states that at a particular sparsity, the best-performing subnetwork has to strike a balance between the effective nodes and paths. However, it does not predict the right balance.
> >
>
> A2: Precisely pinpointing the precise balancing point is indeed a complex challenge. In our research, we initiate by identifying these balancing regions. With the aid of a straightforward proxy method, we endeavor to address this issue, achieving notable empirical balancing and superior performance outcomes. The intricacies of achieving the ideal balance necessitate ongoing exploration, and we believe our work serves as a stepping stone towards this pursuit.
>
> > Q3. It would be interesting to see if this balance depends on sparsity, model, and dataset or model and datasets only. Could the best-performing subnetwork network be maximizing average effective paths per node?
> >
>
> A3: Indeed, the interplay of balance can be influenced by factors such as sparsity, model architecture, and dataset characteristics. Although explicitly pinpointing this balance is intricate and non-trivial, a core contribution of our work is the introduction of an approximation method (NPB) that empirically navigates subnetworks within these balancing regions across diverse sparsities and model types.
> Maximizing average effective paths per node would be an interesting direction. Indeed, a simplistic approach could lead to uniformly distributed connections, resembling the outcomes of Random pruning. To truly achieve a meaningful and effective balance, we believe a more sophisticated strategy is likely necessary.
>
> > Q4. The experiments are limited to image datasets and two neural network architectures. However, I appreciate the authors reporting results over three random seeds though.
> >
>
> A4: Thank you for your suggestions. We would like to clarify that ResNet20 is another version of ResNet which is defined for CIFAR-10/100 tasks. These CIFAR versions are significantly lower in the number of parameters compared to the ImageNet version (ResNet18) and are usually used in pruning literature. Regarding to more network architecture, we have conducted more experiments on two other ResNet versions which are ResNet32 and Wide-ResNet32 on CIFAR-10. These two versions are deeper and wider than ResNet20. We visualize results in Figure 4 in attached file.
>
> > Q5. Shouldn't Eq 4/5 consider the mask M as binary? As such, M is a real value; how is the solution converted to a pruning mask?
> >
>
> A5: We treat M as an integer variable. The optimization problem is solved efficiently via the available convex optimization library (CVXPY library) in which we use the default solver for mixed integer programming named SCIPY [1].
>
> [1] https://www.cvxpy.org/tutorial/advanced/index.html#setting-solver-options
>
> > Q6. Line 262: Should M vary from 0 to hw instead of kw for convolution masks?
> >
>
> A6: Thank you for addressing the error, indeed, M should vary from 0 to hw. We will edit this typo in the next version.

---

> > ### Comment · Reviewer_qyia · 2023-08-17
> >
> > Thanks. Most of my comments are addressed.
> >
> > A1. Thanks, I understand the relevance of NAS benchmarks better now. I would like to highlight that NAS is a data-dependent architecture search and dense, whereas in your case, the search is data-independent and sparse. This could create confusion for the reader --- using this as motivation is ok, but it may be worth addressing this intricate difference in text.
> >
> > A3. While optimizing a ratio may be difficult, it would be nice to visualize the average path per node as a post-hoc analysis to support the argument for optimizing both numbers jointly. And serve as another measure contributing to a good sparsification mask, even though it is harder to optimize. I felt that, given the title of the paper, it appears fitting.

---

> > > ### Author Response · Authors · 2023-08-18
> > > **Response to Reviewer qyia**
> > >
> > > We sincerely appreciate your thoughtful and valuable feedbacks. We will certainly integrate your suggestions in the next version of our paper.

---

### Official Review · Reviewer_8roE · 2023-07-11

**Soundness:** 3 good
**Presentation:** 4 excellent
**Contribution:** 3 good
**Rating:** 6
**Confidence:** 5

**Summary:**

Given the numerous pruning-at-initialization (PaI) methods, the performance of them are still far from satisfactory compared to the post-training pruning methods. In this work, the authors provide a novel perspective to understand the relationship between the performance and the architecture of the subnetworks generated by PaI methods. In particular, this work studies the topology of the subnetwork and observed a positive correlation between the PaI performance and the balance between the effective nodes and effective paths. Further, the authors propose a new PaI method, named Node-Path Balancing Pruner (NPB) to explicitly balance effective nodes and paths. The empirical results demonstrate the effectiveness of the proposed method.

**Strengths:**

1. This paper studies the PaI methods from a novel perspective of model topology. This is a very interesting but also challenging perspective.
2. The figures and graphs in this paper are very delicately plotted and are of high quality. They help the readers better understand the methods proposed.
3. This work made enough literature review, which has covered the most important PaI literature.
4. This work is written in a very coherent manner, which shows clear the motivation, the method, the rationale behind the method, as well as the logic of the empirical studies.
5. The empirical results of the proposed method are indeed very impressive.

**Weaknesses:**

I listed several weaknesses below from different perspectives. I will consider to raise my score if they are properly addressed.
* [Method] Based on my understanding of the NAS observation, the reason for balancing the effective nodes and the effective paths is to maximize the usage of the limited parameter quota. However, if this is true, wouldn't directly cutting down the width of the network a perfect option for NPB? Correct me if I missed anything.
* [Method] Mathematically, the formulation of the optimization objective in Eq. (0) (between Line 243 and Line 244) and Eq. (4~5) are not standard. The to-be-optimized variables should be clearly stated beneath the maximization symbol.
* [Method] It is not very clear to me, how the optimization is carried out through the objective Eq. (4~5). Are there back-propagations involved? It seems not.
* [Experiments] For Fig. 5, the information conveyed is not as clear as I expected. Probably this is because the performance of each method in each setting is annotated with numbers. I would suggest the authors to plot a normal figure showing the performance change (y-axis) vs. the sparsity level (x-axis) as many pruning paper does (there are too many and thus I spare the references here). It can help the readers compare the final accuracy of different methods if they do not care the intermediate results too much (e.g. the effective paths/nodes).
* [Experiments] In the experiments, the $\alpha$ is set to $0.01$. However, the word "balanced" claimed in the abstract seems a bit deceptive. To me, the value of $\alpha$ is very crucial in the proposed method, but the authors fail to demonstrate its role through some ablation studies. I would suggest the authors either explain their choice of $\alpha$ and/or conduct some ablation studies on $\alpha$.
* [Experiments] In Fig. 5, will it be better to add a reference line such as $y = \alpha x$ to help judge the balance of the nodes/paths? Correct me if I missed anything.
* [Experiments] This is not a must but I think the baselines compared in this paper are a bit old. Some methods like ProsPr (already cited in the paper) are recommended to be compared. The lack of this result will not change my evaluation on this paper.
* [Minor] Rephrase: "An effective node/channel is one that at least one effective path goes through it" ==> "A node/channel is effective if at least one effective path goes through it".

**Questions:**

1. It would help the readers to understand the details of the motivation studies if the authors could explain how the "effective sparse ratio" is calculated. Is it done by traversing? Or is there any effective algorithm to calculate that automatically?
2. I am very surprised by the results of the motivation studies. Thus, I am curious if the same rules can be applied to post-training pruning?
3. In the algorithm of NPB, the layer-wise sparsity is obtained through ERK. Is there a particular reason for this choice? Will other methods also be applicable?

**Limitations:**

I do not have additional comments on the limitation of this work. Please refer to the "Weaknesses" and "Questions" sections.

---

> ### Author Rebuttal · Authors · 2023-08-09
>
> We would like to thank you for the time to review our work and are glad that you find our work is well-written with sufficient experiments. We would like to address all the weaknesses pointed out by you point-by-point below:
>
> > Q1. Wouldn't directly cutting down the width of the network a perfect option for NPB?
> >
> A1: We appreciate your engagement with our work. While it's true that balancing effective nodes and paths aims to maximize the utilization of a limited parameter budget, directly cutting down the network's width might not be the ideal solution for pruning tasks.
> Drastically reducing the network's width can lead to a significant drop in representation capacity, potentially causing a performance decline. For example, Synflow, which reduces width to some extent, often sees performance improvements after reshuffling subnetworks. This effect suggests that simply narrowing the network might not fully capture the intricate trade-offs between nodes and paths.
> The balancing of effective nodes and paths is about finding the right equilibrium to maintain an adequate representation capacity while also ensuring information flow. Our proposed approach acknowledges these complexities and seeks to strike a more refined balance to achieve competitive performance within parameter constraints.
> > Q2. The formulation of the optimization objective in Eq. 0,4,5 are not standard.
> >
> A2: We appreciate you pointing out our mistake. We will review and revise the formulations in the updated version.
> > Q3. It is not very clear to me, how the optimization is carried out through the objective Eq. (4~5). Are there back-propagations involved? It seems not.
> >
> A3: Solving node-path balancing objective globally over the whole neural networks seems to be a hard problem, one can conjecture that it is even NP-hard. However, one of our main contributions is to sidestep solving this hard problem by solving a sequence of easy problems to obtain good approximated solutions. In particular, we propose an approximation for solving this problem by doing layer by layer through convex optimization. The approximation problem is solved efficiently via the available convex optimization library (CVXPY library) in which we use the default solver for mixed integer programming named SCIPY [1].
>
> [1] https://www.cvxpy.org/tutorial/advanced/index.html#setting-solver-options
> > Q4. I would suggest the authors to plot a normal figure showing the performance change (y-axis) vs. the sparsity level (x-axis) as many pruning paper does.
> >
> A4: We have already visualized the figure like your suggestion in Appendix F. Please refer to the Appendix F in supplementary for more details.
> > Q5. The choice of $\alpha$ and/or conduct some ablation studies on $\alpha$.
> >
> A5: It's important to mention that optimizing nodes is relatively simpler compared to optimizing paths (we can simply assign parallel edges connecting nodes in the current layer to nodes in the next layer); hence, we deliberately choose small values of alpha, which can be considered as prior knowledge. We set the small alpha ($\alpha = 0.01$) to all settings to ensure fair comparison with other baselines. In Appendix G, we have presented an ablation study on $\alpha$ and $\beta$.
> > Q6. In Fig. 5, will it be better to add a reference line such as $y=ax$ to help judge the balance of the nodes/paths?
> >
> A6: It is challenging to identify the specific balancing line. Balancing between node and path is indeed a complex endeavor and non-trivial due to their mutual impact on each other. We propose a practical proxy method to address the complex balancing issue which simply offers strong empirical balancing results across a range of experiments. Our goal with the proxy method is to offer a practical solution that strikes a balance while being easily applicable.
> > Q7. Comparison with recent baselines such as ProsPr
> >
> A7: We appreciate your recommendation and understand your perspective on including more recent baselines. While ProsPr belongs to PaI approach, it's worth noting that ProsPr employs multiple gradient descent steps to identify important weights. Thus, we believe that ProsPr has certain advantages over PaI methods like our NPB (which does not use information from the dataset).
> > Q8. [Minor] Rephrase: "An effective node/channel is one that at least one effective path goes through it" ==> "A node/channel is effective if at least one effective path goes through it"
> >
> A8: Thank you for your suggestions, we will edit it in the next version.
> > Q9. How the "effective sparse ratio" is calculated.
> >
> A9: We follow the algorithm in Appendix M of [2], The effective sparse ratio = #Effective parameters / #Unpruned parameters
>
> We determine effective parameters as follow:
>
> 1. Set each weight in the network to 1 if it is unpruned or 0 if it is pruned.
> 2. Forward-propagate a single example comprising all 1’s.
> 3. Compute the sum of the logits.
> 4. Compute the gradients with respect to this sum.
> 5. Prune any unpruned weight with a gradient of 0. Since these weights did not receive any gradient, they are disconnected from the output of the network.
>
> [2] https://arxiv.org/abs/2009.08576
> > Q10. I am curious if the same rules can be applied to post-training pruning?
> >
> A10: We consider LTH as a representative method for post-training pruning approach. Through the additional experiments shown in Figure 1 in the attached file, we observe that LTH’s subnetworks are also in the same rules. The gap in performance between LTH and PaI methods is due to LTH leveraged information from the dataset and model training to produce subnetworks. Overall, we strongly believe these results align with our principle and further support our principle as a necessary condition for good PaI.
> > Q11. Why choose ERK to obtain layer-wise sparsity? Will other methods also be applicable?
> >
> A11: We have mentioned this ablation study in Appendix G. Due to limited 6000 words, we refer reviewer to Appendix G in the supplementary for more details.

---

> > ### Comment · Reviewer_8roE · 2023-08-19
> > **Thank you for your response.**
> >
> > Dear authors,
> >
> > Thank you for your response. I have carefully read your response and my concerns have mostly been alleviated. Therefore, I decide to keep my original rating of 6 and will vote for acceptance if AC asks. Thank you again for your hard work.
> >
> > Best,

---

> > > ### Author Response · Authors · 2023-08-19
> > > **Response to Reviewer 8roE**
> > >
> > > We would like to thank you again for your valuable feedbacks and insights. We will certainly integrate your suggestions in the next version of our paper.

---

### Official Review · Reviewer_a28p · 2023-07-18

**Soundness:** 2 fair
**Presentation:** 2 fair
**Contribution:** 2 fair
**Rating:** 5
**Confidence:** 4

**Summary:**

The authors propose a Pruning at Initialization (PaI) method that considers the balance between the number of effective nodes and effective paths. This design principle is based on the observations on the NAS benchmark as well as layer-wise reshuffling. The pruning problem is nicely formulated as a multi-objective optimization problem. The experiment results show that the proposed method NPB outperforms the state-of-the-art method PHEW in some configurations.

**Strengths:**

* A new Pruning at Initialization (PaI) method is developed considering both the number of effective nodes and the number of effective paths.
* A lot of analyses and experiments have been conducted to show the superiority and motivation of the proposed method.
* The pruning problem is formally transformed into a multi-objective optimization problem.
* In some settings, the proposed method NPB is shown to outperform the state-of-the-art method PHEW.

**Weaknesses:**

The motivation and technical details are not very clear in several places, as follows.

**Overall:**
* The sweet spot of the proposed framework seems to be the extreme sparsity regime (> 99%).  I am really wondering if we need to prune a network to the extreme. My impression is that the sweet spot is a corner case.

**Section 3.3:**
* Line 187: Why does reshuffling make subnetworks wider?
* Line 205: How is the hypothesis derived from reshuffling?  PaI actually does *not* involve reshuffling.

**Section 4.1:**
* Figure 4 (and Figure 1): It seems that, in typical architectures, the number of effective nodes and the number of effective paths are strongly correlated.  Then, why do we need to balance these two metrics?
* Line 223: Along the same lines, the strong correlation between the two metrics indicates that one of them is redundant. Thus, the overall claim cannot be supported by the observation.
* Figure 4 (and Figure 1): What's the meaning of each value in the x- or y-axis?  For example, what do you mean by $9.0$ in the x-axis?

**Section 4.2:**
* Due to the strong correlation between the number of effective nodes and the number of effective paths, it may be unnatural to produce subnetworks with too many effective paths (or nodes) and too few effective nodes (or paths).

**Section 4.3:**
* Even though it is reasonable to balance these two metrics, their ranges significantly differ. Actually, they have completely different scales (natures). This *incompatibility* between the two metrics would explain a very small value of $\alpha$=0.01. Syntactically, Equation (4) does not achieve the balance, but mainly considers only $f_p$. Also, $\alpha=0.01$ is indeed arbitrary and does not provide any insight on the optimal balance.
* Appendix G (Ablation Study): Figure 8 shows that the accuracy is not that sensitive to $\alpha$. I am still very confused why the balance between the two metrics is indeed important. Figure 8 directly shows that the optimal balance may not be important.

**Section 5.2:**
* It is not clear how the sparsity levels were chosen for the datasets in Figure 5. That is, how are the 12 settings chosen? I believe that a wider range of sparsity levels should be analyzed.
* In contrast to Figure 4 (and Figure 1), it is not clear why most of the circles are not placed on the diagonal.
* Line 316: Many cases of NPB do not lie in the so-called balancing regions. It would be better to mark the balancing regions in the figures.

Overall, I believe the proposed framework is interesting and has some potential. However, the main contribution is neither well motivated nor rigorously presented. It seems that my concerns may not be sufficiently addressed by the rebuttal process. Therefore, I would like to give my rating as a reject.

--

**After Rebuttal**

Some of the concerns and questions are resolved by the authors' rebuttal. However, the observational motivation and the need for careful optimization are not convincingly presented. Thus, I am a little reluctant to give a high score and would like to increase my rating to 5.

**Questions:**

See the weak points mentioned above.

**Limitations:**

The authors did not discuss the limitations and potential negative societal impact.

---

> ### Author Rebuttal · Authors · 2023-08-09
>
> Thank you for your time to review our work. We would like to address all the weaknesses pointed out by you point-by-point below:
> > The sweet spot of the proposed framework seems to be the extreme sparsity regime (> 99%)
> >
> A1: We respectfully disagree with the reviewer. Our experiments have been conducted across a range of sparsity levels, all of which are lower than 99%. The results demonstrate superior performance compared to baseline methods.
> > Why does reshuffling make subnetworks wider?
> >
> A2: Iterative pruning prioritizes high-scoring nodes, which are typically determined by gradient/weight information. When nodes have edges pruned, their scores tend to decrease, making them more susceptible to further pruning. This can lead to slender subnetworks (eg., SynFlow, Iter-SNIP). When shuffling layer-wise, connections are redistributed uniformly, widening the subnetwork. Similar findings have been illustrated in [1].
>
> [1] https://arxiv.org/abs/2009.08576
> > How is the hypothesis derived from reshuffling? PaI actually actually does *not* involve reshuffling.
> >
> A3: In general, PaI approaches do not involve reshuffling. However, we use reshuffling to emphasize the balance between effective nodes and paths. Good sparse NNs should have enough effective nodes to ensure good representation capacity and enough effective paths to guarantee good signal propagation during training. With limited number of parameters, shuffling can intuitively widen subnetworks, but it can decrease the number of effective paths, which can impede information flow (good signal path can be destroyed after shuffling). The empirical results perfectly align with our intuition: While shuffling can sometimes boost performance, accuracy drops as sparsity increases. Stemming from the above observation, we propose to balance two metrics, enhancing optimal sparse NNs design.
> > Figures 1, 4: in typical architectures, the number of effective nodes and paths are strongly correlated, which indicates that one of them is redundant.
> >
> A4: We disagree with the reviewer for two reasons:
>
> - In NAS, networks are dense. Correlations between nodes and paths exist depending on specific design aspects like the number of channels. However, components like skip-connect or pooling layers may affect either nodes or paths. Different block configurations (Appendix H) results in variable node and path counts.
> - Node-path relationship is totally different in pruning. All subnetworks are pruned from the same dense network, which have a strong constraint of the fixed number of connections. The relationship is now based on the remaining connections. Therefore, the way we prune the network can significantly affect this relationship.
>
> > Then, why do we need to balance these two metrics? Thus, the overall claim cannot be supported by the observation.
> >
> A5: It's important to emphasize that in typical architectures, nodes and paths are related and used to monitor the network's performance.
> A notable node-path balancing scenario is preventing uniformly distributing connections to nodes to enhance the representation capacity (Random pruning), which implicitly limits information flow.
> Reversibly, the node-path balancing also prevents pruning out most of the channels but greatly keeps the connections between nodes (SynFlow), which makes networks have a poor number of linear-regions to distinguish features.
> The node-path mechanism aids us in identifying the optimal subnetwork that is determined by both metrics. This approach results in superior performance compared to the use of a single metric as we have shown in our paper.
> > Figures 1, 4: What's the meaning of each value in the x- or y-axis?
> >
> A6: The x- and y-axis are the number of effective nodes (in ln scale) and paths (in log scale). We mentioned these numbers in lines 284-286.
> > Due to the strong correlation between nodes and paths, it may be unnatural to produce subnetworks with too many paths (or nodes) and too few nodes (or paths).
> >
> A7: In pruning as explained in A4, it is natural that we can make the network slender with a many paths (like Synflow) or broader with fewer paths (like PHEW) as shown in Figure 5.
> > Different scales of metrics in Eq-4. $\alpha=0.01$  is indeed arbitrary and does not provide any insight on the optimal balance.
> >
> A8: Due to limited space, we refer Reviewer to lines 269-274 for details.
> > Appendix G, Figure 8 shows that the accuracy is not that sensitive to $\alpha$ and optimal balance may not be important.
> >
> A9: In Figure 8, with low sparsity levels (<90%), optimizing node is easy because there are many connections, and differences in $\alpha$ only affect the number of paths. However, as sparsity increases and $\alpha$ varies, the limited number of weight directly impacts the number of nodes in subnetworks. By fixing $\beta$ and varying $\alpha$, subnetworks’ performance changes accordingly. Thus, identifying an optimal balance point is crucial for pruning neural networks.
> > How sparsity levels were chosen? Wider range of sparsity should be analyzed
> >
> A10: The choice of sparsities was guided by the compression rate r which is computed as follows #Remaining_weight / #All_weight = 10^(-r) [2]. Sparsity levels are derived from $r\in$ {0.5, 1, 1.5, 2}. While wider sparse range is important, we believe that the chosen sparsity settings are common and effectively demonstrate the efficiency of our proposed method.
>
> [2] https://arxiv.org/abs/2006.05467
> > In contrast to Figures 1, 4, it is not clear why most of the circles are not placed on the diagonal.
> >
> A11: Please refer to A4 and A7.
> > Many cases of NPB do not lie in the so-called balancing regions. It would be better to mark the balancing regions in the figures.
> >
> A12: We recognize the need to clarify effective nodes and paths' balancing regions. Balancing these factors is complex due to their mutual impact. Our proposed proxy method addresses this challenge, yielding strong empirical balancing outcomes across various experiments.

---

> > ### Comment · Reviewer_a28p · 2023-08-11
> > **Response to Authors**
> >
> > Thank you very muuch for the authors' detailed responses. Some of my concerns and questions are resolved. However, the observational motivation and the need for careful optimization are not still clear. Overall, I would like to increase my rating to 5. The below is my response to each of the answers.
> >
> > * A1: Line 54 and 191, the authors explained the motivation using the >99% regime.
> > * A2: Thanks for your explanation.
> > * A3: Makes sense.
> > * A4: If so, why is Section 4.1 (NAS Observations) relevant to your claim. This is still not clear.
> > * A5: Understood, but my point was connection from Section 4.1 to Section 4.2, which is still not clear.
> > * A6: Better to specify the base in the figures.
> > * A7: Okay.
> > * A8: I read it during the review, but it was too vague.
> > * A9: Even when $Sparisty$=96.84, optimizing $\alpha$ does not seem to be important. When $\alpha$ increased 90 times (from 0.01 to 0.9) at $\beta$ = 1.0, the accuracy changed by only 3% (and even increased, as opposed to A8). Thus, it is not still clear why optimizing $\alpha$ is crucial.
> > * A10: Understood.
> > * A11: Understood.
> > * A12: Hope to improve the presentation.

---

> > > ### Author Response · Authors · 2023-08-13
> > > **Response to Reviewer a28p**
> > >
> > > We really thank you for your careful read and detailed point-by-point response to our rebuttal, we truly appreciate it! We would like to add some further clarifications as below:
> > >
> > > - In the shuffling experiment, we show a counter-argument against prior findings [1] which claim that PaI methods are insensitive in random shuffling. We indicate that when sparsity level increases (especially >99%) reshuffling significantly reduces the performance of subnetworks. We also provide a potential (and empirically observed) explanation to this phenomenon which is the significant decrease in the number of effective paths after reshuffling. This observation leads to our idea of node path balancing.
> > >
> > > - The motivation behind the NAS observation is that, in the NAS experiment, we observe a general trend that networks with better performance tend to have a high number of effective paths and nodes together. This, together with the reshuffling observation, makes us believe that the node path balancing hypothesis/principle is essential, which motivates the rest of our work. In particular, we conjecture that the similar trend should hold for sparse NNs. To evaluate this, we decided to apply weight reshuffling on SNIP and Synflow, as reshuffling weights after pruning will intuitively destroy the good information flow (similar as effective path) that is preserved by these methods. As we know that SNIP and Synflow tend to prune weights with smallest impact of the synaptic flow. Our results are in line with our hypothesis: when sparsity increases, reshuffling layer-wise significantly impedes information flow with lower number of effective paths (though with more effective nodes), damaging the performance of these methods. Therefore, our preliminary experiments demonstrate that maintaining the good balance of effective nodes and paths are crucial for sparse NNs.
> > >
> > > - Regarding choosing $\alpha$, our objective in Eq. 4~5 is to maximize simultaneously both effective nodes and paths. However, the relationship between the number of effective nodes and paths is complicated and depends on various aspects such as architecture, sparsity level etc. In some cases, increasing nodes (paths) leads to reduction in paths (nodes). Therefore, $\alpha$ is considered an adjusted term between two metrics. Choosing exactly alpha or beta is non-trivial, and it is sensitive to the subnetwork’s performance. As the reviewer pointed out, “when $\alpha$ increased 90 times (from 0.01 to 0.9) at $\beta = 1$ the accuracy changed by only 3%”. But we argue that this change is ineligible especially in the pruning context. As shown in Figure 8, if we can choose a good pair of alpha and beta, we will have superior performance subnetworks.
> > >
> > > - We will make the above motivation much clearer in the camera-ready version.
> > >
> > > [1] https://arxiv.org/abs/2009.08576

---

> > > > ### Comment · Reviewer_a28p · 2023-08-15
> > > > **Response to Authors**
> > > >
> > > > I deeply appreciate the authors' detailed response.  However, this response repeats the previous claims, which I believe to be inconsistent, incomplete, or incorrect.
> > > > * The motivation does not cover the entire range of sparsity, but only covers the >99% regime, even though the accuracy is claimed to be good in the entire range.
> > > > * It would be much better to motivate the node-path balancing principle directly using *sparse* NNs instead of dense NNs in Section 4.1.
> > > > * It seems that the users need to *exhaustively* search for the optimal values of $\alpha$ and $\beta$.  Some pairs of $\alpha$ and $\beta$ produce the accuracy worse than PHEW.  The optimization objective needs to be improved in this context as future work.

---

> > > > > ### Author Response · Authors · 2023-08-15
> > > > > **Response to Reviewer a28p**
> > > > >
> > > > > Apologies for still having unclear explanations in our previous clarification attempt. Here’s our response to your latest comments:
> > > > >
> > > > > >**The motivation does not cover the entire range of sparsity, but only covers the >99% regime, even though the accuracy is claimed to be good in the entire range.**
> > > > > >
> > > > >
> > > > > We believe there is indeed a misunderstanding here. We used shuffling as a motivation because we wanted to find a tool that can explain why shuffling worked in certain cases while failed in other ones (i.e., <99% and >99% sparsity regimes, respectively) – and NPB turned out to be this tool we were looking for.
> > > > >
> > > > > In more detail, we started doing experiments with shuffling to better understand it, after reading the paper of Frankle et al. (2021). During these experiments, we realised that shuffling is not always as good as claimed in the original paper. Instead, what we saw is that from sparsity levels > 99% onwards, the performance starts dropping significantly after shuffling. This was a surprising result, and we didn’t understand why (we believe this finding is also significant itself and is of independent interest). When we extensively investigated the details, we actually identified the reason for this new behaviour to be the significant drop in the number of paths after shuffling (i.e., they were not well-balanced) which substantially impedes information flow. Besides, we’ve also found that at normal sparsity levels (< 99%), shuffling would broaden the width of the subnetworks, which in turn can improve the performance due to enhancing representation capacity. Note that both phenomena can be explained via the NPB principle. These observations, together with the NAS experiments, helped us form the idea of explicitly balancing nodes and paths, which we believed to be essential. That's how we decided to design a new algorithm which explicitly focuses on finding an efficient balance.
> > > > >
> > > > > As such, we’ve decided to use the shuffling topic to demonstrate that the lack of this node path balance eventually leads to the performance drop of shuffling in the >99% sparsity regime, or due to achieving a better balance, shuffling can achieve accuracy improvement in the <99% sparsity regime (as with SynFlow, SNIP). Section 3.3 was meant to highlight this unforeseen performance changes which requires further explanation, which then leads to the need for introducing the NPB principle as a possible explanation. In particular, we first highlighted the >99% sparsity regime where the new phenomenon of performance drop occurs, and not because our NPB works well in this regime here only (lines 185-190). We also discussed the <99% regime where we explained via the lens of NPB why shuffling improves performance of SNIP and SynFlow (lines 191-197). Note that this performance improvement after shuffling was also observed by Frankle et al. (2021).
> > > > >
> > > > > Hence, the motivation of using shuffling is to provide an application (i.e., the shuffling mechanism) where the surprising performance changes in both normal and extreme regimes can be explained by using the NPB principle. Notice that we mentioned this motivation in more detail in Section 1 (lines 52-63).
> > > > >
> > > > > We hope this explanation provides a more thorough clarification. We also understand that in the current version our paper might have not conveyed this message clearly and therefore created confusions. We will improve it in the next version.
> > > > >
> > > > > >**It would be much better to motivate the node-path balancing principle directly using sparse NNs instead of dense NNs in Section 4.1.**
> > > > > >
> > > > >
> > > > > You are correct. It would have been much better to have a ready-to-go and comprehensive dataset of different sparse NNs (which does not exist to date unfortunately). However, we believe that the NAS-Bench-Macro dataset is already good enough to provide some hints on the commonality of good network architectures, as the NPB principle does not only hold for sparse networks, but we believe for dense networks as well.
> > > > >
> > > > > In more detail, we wanted to identify some ideas on how to generate network topologies with potentially good performance. To do so, we’ve looked at the NAS-Bench-Macro dataset to investigate whether NNs with good performance have somthing in common. Our focus was mainly on the topological insights of these networks. As a result, we’ve found that good networks tend to have some balance between their numbers of nodes and paths. This is how we started forming the conjecture about the NPB principle.
> > > > >
> > > > > >**It seems that the users need to exhaustively search for the optimal values of \alpha and \beta. Some pairs of \alpha and \beta produce the accuracy worse than PHEW. The optimization objective needs to be improved in this context as future work.**
> > > > > >
> > > > >
> > > > > You are absolutely correct here. We do not claim that the work is done and that our optimization approach is perfect. There is still definitely room for future work to derive more principled optimisation models to achieve better balance points.

---

> > > > > > ### Comment · Reviewer_a28p · 2023-08-18
> > > > > >
> > > > > > Thank you for your detailed response. Now, I have a better understanding on your story. I think the motivation part needs a major revision, and the current limitation (especially, on the optimization model) should be clearly stated. Please try to improve the manuscript as much as possible if accepted.

---

> > > > > > > ### Author Response · Authors · 2023-08-18
> > > > > > > **Response to Reviewer a28p**
> > > > > > >
> > > > > > > We sincerely appreciate your thoughtful and valuable insights and discussion, which have highlighted the vagueness in the motivation part of our paper. We will certainly incorporate those clarifications accordingly into the next version of our paper.

---

### Author Rebuttal · Authors · 2023-08-09

Thank you for your valuable and constructive feedbacks. We have performed the additional experiments as requested by the reviewers and have provided the results in this pdf file.
We hope that our responses address your concern. If you have any additional questions, uncertainties, or areas you would like us to elaborate on, we are happy to engage in a continued discussion and provide any additional information you may require.

---

### Decision · Program_Chairs · 2023-09-21

**Decision:**

Accept (poster)

**Comment:**

The paper provides a technique for pruning networks at initialization (PaI). The method provided relies on an observation that there exists a correlation between the performance of a network and the balance between the effective nodes and effective path. The reviews agree that this approach is novel, and furthermore, that the main hypothesis is explained well along with convincing arguments and experiments for its correctness. Given the large amount of works dealing with pruning, having a novel technique makes this paper appealing. The main concern raised for the paper is the quality of its writing. This was explicitly mentioned in some reviews and implicitly in others that had many clarification questions. This being said, the required edits are listed in the discussion here, and the task of integrating these changes towards a final version seems feasible to me. Given this and the mentioned strengths of the paper, I recommend its acceptance.